# Through the Dual-Prism: A Spectral Perspective on Graph Data Augmentation for Graph Classification

## Abstract

Graph Neural Networks (GNNs) have become the preferred tool to process graph data, with their efficacy being boosted through graph data augmentation techniques. Despite the evolution of augmentation methods, issues like graph property distortions and restricted structural changes persist. This leads to the question: *Is it possible to develop more property-conserving and structure-sensitive augmentation methods?* Through a spectral lens, we investigate the interplay between graph properties, their augmentation, and their spectral behavior, and found that keeping the low-frequency eigenvalues unchanged can preserve the critical properties at a large scale when generating augmented graphs. These observations inform our introduction of the Dual-Prism (DP) augmentation method, comprising DP-Noise and DP-Mask, which adeptly retains essential graph properties while diversifying augmented graphs. Extensive experiments validate the efficiency of our approach, providing a new and promising direction for graph data augmentation.

## 1 Introduction

Graph structures, modeling complex systems through nodes and edges, are ubiquitous across various domains, including social networks (Newman et al., 2002), bioinformatics (Yi et al., 2022), and transportation systems (Jin et al., 2023a). Graph Neural Networks (GNNs) (Kipf & Welling, 2016a) elegantly handle this relational information, paving the way for tasks such as accurate predictions. Their capabilities are further enhanced by graph data augmentation techniques. These methods artificially diversify the dataset through strategic manipulations, thereby bolstering the performance and generalization of GNNs (Rong et al., 2019; Feng et al., 2020; You et al., 2020). Graph data augmentation has progressed from early random topological modifications, exemplified by DropEdge (Rong et al., 2019) and DropNode (Feng et al., 2020), to sophisticated learning-centric approaches like InfoMin (Suresh et al., 2021). Furthermore, techniques inspired by image augmentation's mixup principle (Zhang et al., 2017) have emerged as prominent contenders in this domain (Verma et al., 2019; Wang et al., 2021; Guo & Mao, 2021).

Though promising, these augmentation methods are challenged by three key issues as follows. (1) *Graph Property Distortion.* Before the era of deep learning, graph properties, e.g., graph connectivity and diameter, served as vital features for classification for decades (Childs et al., 2009). While now they seem to be ignored, many aforementioned contemporary augmentation methods appear to sidestep this tradition and overlook the graph properties. For instance, an example graph from the IMDB-BINARY dataset (Morris et al., 2020) and its augmented graph via DropEdge are illustrated in Figures 1a and 1b, respectively. The polar plot in Figure 1e shows the properties of these graphs, where each axis represents a distinct property. It is evident that DropEdge significantly alters the original graph's properties, as indicated by the stark difference between the shapes of the orange (original) and blue (augmented) pentagons. (2) *Limited Structural Impact.* The majority of existing methods' localized alterations do not capture the broader relationships and structures within the graph, limiting their utility. Consider a social network graph, where removing an edge affects just the immediate node and does little to alter the overall community structure. We thus ask: Can we design more *property-retentive* and *structure-aware* data augmentation techniques for GNNs?

| (a) Original Graph | (b) DropEdge | (c) DP-Noise | (d) DP-Mask | (e) Property Change | (f) Spectral Change |

Figure 1: Visualization of (a) a graph from the IMDB-BINARY dataset and its augmented graphs via (b) DropEdge (Rong et al., 2019), (c) DP-Noise (ours), and (d) DP-Mask (ours). Dashed line: Dropped edge. Red line: Added edge. (e) Five properties of these graphs. $r$: radius. $d$: diameter. conn.: connectivity. ASPL: average shortest path length. #peri: number of periphery. Ori.: Original. D.E.: DropEdge. DP-N: DP-Noise. DP-M: DP-Mask. (f) The eigenvalues of these four graphs.

**Through the Dual-Prism: A Spectral Lens.** Graph data augmentation involves altering components of an original graph. These modifications, in turn, lead to changes in the graph's spectral frequencies (Ortega et al., 2018). Recent research highlighted the importance of the graph spectrum: it can reveal critical graph properties, e.g., connectivity and radius (Chung, 1997; Lee et al., 2014). Additionally, it also provides a holistic summary of a graph's intrinsic structure (Chang et al., 2021), providing a global view for graph topology alterations. Building on this foundation, a pivotal question arises: *Could the spectral domain be the stage for structure-aware and property-retentive augmentation efforts?* Drawing inspiration from *dual prisms*—which filter and reconstruct light based on spectral elements—can we design a *polarizer* to shed new light on this challenge? With this in mind, we use spectral graph theory, aiming to answer the following questions: 1) Can a spectral approach to graph data augmentation preserve essential graph properties effectively? 2) How does spectral-based augmentation impact broader graph structures? 3) How does spectral-based augmentation compare to existing methods in enhancing the efficiency of GNNs for graph classification?

We begin with an empirical exploration in Section 3, where we aim to understand the interplay between topological modifications and their spectral responses. Our insights reveal that changes in graph properties mainly manifest in *low-frequency components*. Armed with this, in Section 4, we unveil our Dual-Prism (DP) augmentation strategies, DP-Noise and DP-Mask, by only changing the high-frequency part of the spectrum of graphs. Figures 1c and 1d provide a visualization of the augmented graphs via our proposed methods, i.e., DP-Noise and DP-Mask. As shown in Figure 1e, compared with DropEdge, our approaches skillfully maintain the inherent properties of the original graph, differing only slightly in the ASPL. Note that although we solely present one example underscoring our method's capability, its robustness is consistently evident across all scenarios.

In addition to the properties, we further explore the spectrum comparison, shown in Figure 1f. Compared with DropEdge, the spectrum shifts caused by our methods are noticeably smaller. Interestingly, despite our approaches' relative stability in the spectral domain, they induce substantial changes in the spatial realm (i.e., notable edge modifications). This spectral stability helps retain the core properties, while the spatial variations ensure a rich diversity in augmented graphs. Conversely, DropEdge, despite only causing certain edge changes, disrupts the spectrum and essential graph properties significantly. Simply put, **our methods skillfully maintain graph properties while also diversifying augmented graphs.** In Section 5, we evaluate the efficacy of our methods on graph classification, across diverse settings: supervised, semi-supervised, unsupervised, and transfer learning on various real-world datasets. Our concluding thoughts are presented in Section 6.

**Contributions.** Our main contributions are outlined as follows. *(1) Prism – Bridging Spatial and Spectral Domains:* We introduce a spectral lens to shed light on spatial graph data augmentation, aiming to better understand the spectral behavior of graph modifications and their interplay with inherent graph properties. *(2) Polarizer – Innovative Augmentation Method:* We propose the globally-aware and property-retentive augmentation methods, Dual-Prism (DP), including DP-Noise and DP-Mask. Our methods are able to preserve inherent graph properties while simultaneously enhancing the diversity of augmented graphs. *(3) New Light – Extensive Evaluations:* We conduct comprehensive experiments spanning supervised, semi-supervised, unsupervised, and transfer learning paradigms on 21 real-world datasets. The experimental results demonstrate that our proposed methods can achieve state-of-art performance on the majority of datasets.

## 2 RELATED WORK

**Spectrum and GNNs.** Spectral graph theory (Chung, 1997) has found an appealing intersection with GNNs (Ortega et al., 2018; Wu et al., 2019; Dong et al., 2020; Bo et al., 2021; Chang et al., 2021; Yang et al., 2022). Early GNN approaches employed the spectrum of the Laplacian matrix to define convolution operations on graphs in the spectral domain (Hammond et al., 2011; Defferrard et al., 2016). As it evolved, there was a strategic shift towards spectral methods to enhance scalability (Nt & Maehara, 2019). This spectral perspective continues to be influential across various graph learning domains, notably in graph contrastive learning (GCL) (Liu et al., 2022; Lin et al., 2022), adversarial attacks (Entezari et al., 2020; Chang et al., 2021), and multivariate time series learning (Cao et al., 2020; Jin et al., 2023b). Zooming into the GCL domain, where data augmentation plays a pivotal role, Liu et al. (2022) introduced the general rule of effective augmented graphs in GCL via a spectral perspective. Lin et al. (2022) presented a novel augmentation method for GCL, focusing on the invariance of graph representation in the spectral domain. Notably, while these studies offer valuable insights, they mainly concentrate on the GCL paradigm, often neglecting broader discussions about preserving the core properties of graphs and supervised tasks.

**Data Augmentations for GNNs.** Graph data augmentation refers to the process of modifying a graph to enhance or diversify the information contained within, which can be used to bolster the training dataset for better generalization or model variations in real-world networks Ding et al. (2022); Zhao et al. (2022). Early methods are grounded in random modification to the graph topology. Techniques like DropEdge (Rong et al., 2019), DropNode (Feng et al., 2020), and random subgraph sampling (You et al., 2020) introduce stochastic perturbations in the graph structure. In addition to random modification, there is a wave of methods utilizing more sophisticated, learning-based strategies to generate augmented graphs (Suresh et al., 2021). Another research line is inspired by the efficiency of mixup (Zhang et al., 2017) in image augmentation, blending node features or entire subgraphs to create hybrid graph structures (Verma et al., 2019; Wang et al., 2021; Guo & Mao, 2021; Han et al., 2022; Park et al., 2022; Ling et al., 2023). However, while the above techniques, either spatial or spectral, have advanced the field of graph data augmentation, challenges remain, especially in preserving broader structural changes and graph semantics. Our work presents a new spectral augmentation approach, providing a principled way to modulate a graph's spectrum while preserving the core, low-frequency patterns.

## 3 A SPECTRAL LENS ON GRAPH DATA AUGMENTATIONS

### 3.1 PRELIMINARIES

An undirected graph $\mathcal{G}$ is represented as $\mathcal{G} = (V, E)$ where $V$ is the set of nodes with $|V| = N$ and $E \subseteq V \times V$ is the set of edges. Let $A \in \mathbb{R}^{N \times N}$ be the adjacency matrix of $\mathcal{G}$, with elements $a_{ij} = 1$ if there is an edge between nodes $i$ and $j$, and $a_{ij} = 0$ otherwise. Let $D \in \mathbb{R}^{N \times N}$ be the degree matrix, which is a diagonal matrix with elements $d_{ii} = \sum_j a_{ij}$, representing the degree of node $i$. The Laplacian matrix of $\mathcal{G}$ is denoted as $L = D - A \in \mathbb{R}^{N \times N}$. The eigen-decomposition of $L$ is denoted as $U \Lambda U^\top$, where $\Lambda = diag(\lambda_1, \ldots, \lambda_N)$ and $U = [u_1^\top, \ldots, u_N^\top] \in \mathbb{R}^{N \times N}$. For graph $\mathcal{G}$, $L$ has $n$ non-negative real eigenvalues $0 \leq \lambda_1 \leq \lambda_2 \leq \ldots \leq \lambda_N$. Specifically, the *low-frequency components* refer to the eigenvalues closer to 0, and the *high-frequency components* refer to the relatively larger eigenvalues.

### 3.2 SPECTRAL ANALYSIS INSIGHTS

Adopting a spectral viewpoint, we conduct a thorough empirical study to understand the interplay between graph properties, graph topology alterations in the spatial domain, and their corresponding impacts in the spectral realm. The findings from our analysis include three crucial aspects as detailed below. Details of experimental settings and more results can be found in Appendices C & D.

**Obs 1. The position of the edge flip influences the magnitude of spectral changes.**

In Figures 2a and 2b, we explore how adding different edges to a toy graph affects its eigenvalues. For instance, the addition of the edge 1↔3 (shown as the red line), which connects two proximate nodes, primarily impacts the high-frequency component $\lambda_6$ (highlighted by the red rectangle). In contrast, when adding edge 2↔6 (the blue line) between two distant nodes, the low-frequency component $\lambda_1$ exhibits the most noticeable change (indicated by the blue rectangle). These variations in the spectrum underscore the significance of the edge-flipping position within the graph's overall

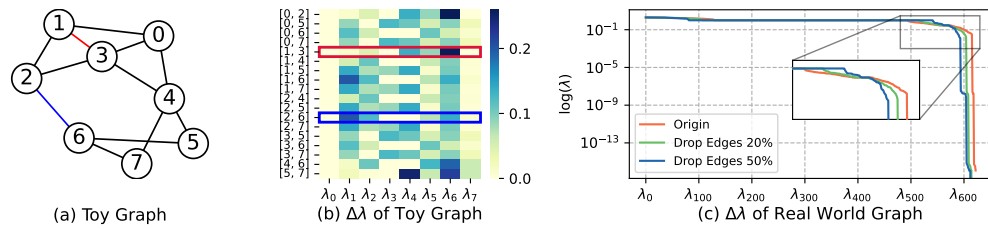

Figure 2: (a) A toy graph $\mathcal{G}$ consisting of eight nodes. (b) Absolute variation in eigenvalues of $\mathcal{G}$ when adding an edge at diverse positions. The red and blue rectangles represent when adding the corresponding edges in $\mathcal{G}$ and the change of the eigenvalues. (c) A real-world case in the REDDIT-BINARY dataset where, when dropping 20% and 50%, the high frequency is more vulnerable.

topology, whose insight is consistent with findings by (Entezari et al., 2020; Chang et al., 2021). Such spectral changes not only affect the graph's inherent structural features but also potentially affect the outcomes of tasks relying on spectral properties, such as graph-based learning.

**Obs 2. Low-frequency components display greater resilience to edge alterations.**

Building on Obs 1, we further investigate the phenomenon of different responses of high- and low-frequency components to topology alterations using a real-world graph from the REDDIT-BINARY dataset. We apply DropEdge (Rong et al., 2019) for data augmentation by first randomly dropping 20% and 50% of edges and then computing the corresponding eigenvalues, as depicted in Figure 2c. Our findings indicate that, under random edge removal, low-frequency components exhibit greater robustness compared to their high-frequency counterparts.

**Obs 3. Graph properties are crucial for graph classification.**

Certain fundamental properties of graphs, e.g., diameter and radius, are critical for a variety of downstream tasks, including graph classification (Feragen et al., 2013). In Figures 3a and 3b, we present the distributions of two key graph properties – diameter $d$ and radius $r$ – across the two classes in the REDD-M12 dataset. The different variations in these distributions emphasize their critical role in graph classification. Nevertheless, arbitrary modifications to the graph's topology, e.g., random-manner-based augmentation techniques, could potentially distort these vital properties, illustrated in Figures 1b and 1e.

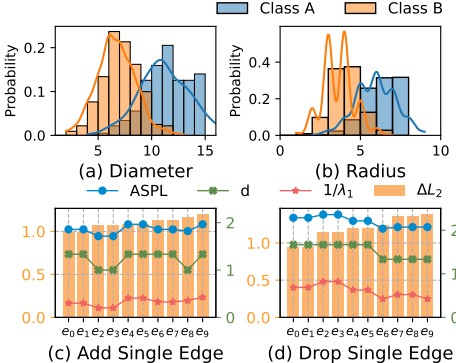

Figure 3: (a) Diameter and (b) radius distributions of different classes in REDD-M12. When (c) adding or (d) removing an edge, variation of the spectral domain $\Delta L_2$, $1/\lambda_1$ of $\mathcal{G}'$, ASPL and diameter $d$ of $\mathcal{G}'$.

**Obs 4. Specific low-frequency eigenvalues are closely tied to crucial graph properties.** From Obs 3, a question is raised: *Can we retain the integrity of these essential graph properties during augmentation?* To explore this, we turn our attention back to the toy graph in Figure 2a and examine the evolution of its properties and eigenvalues in response to single-edge flips. In Figures 3c and 3d, we chart the graph's average shortest path length (denoted by blue dots), diameter $d$ (denoted by green dots) against the reciprocal of its second smallest eigenvalue $1/\lambda_1$ (denoted by red dots)[1]. Our observations reveal a notable correlation between the alterations in $d$ and $1/\lambda_1$.

Further, we investigate correlations among overall spectral shifts, graph properties, and specific eigenvalues. Consistent with established methodologies (Lin et al., 2022; Wang et al., 2022; Wills & Meyer, 2020), we adopt the Frobenius distance to quantify the overall spectral variations by computing the $L_2$ distance between the spectrum of $\mathcal{G}$ and augmented graph $\mathcal{G}'$. Notably, this spectral shift does not directly correspond with the changes in properties or eigenvalues. This suggests that by maintaining critical eigenvalues, primarily the low-frequency components, we can inject relatively large spectral changes without affecting essential graph properties. This observation thus leads to our proposition: *preserving key eigenvalues while modifying others enables the generation of augmented graphs that uphold foundational properties*, instead of only focusing on the overall spectral shifts.

---

[1]For visual clarity, we scaled $d$ and $1/\lambda_1$ by dividing it by 3 and 5, respectively.

## 4 METHODOLOGY

Drawing inspiration from how prisms decompose and reconstruct light and how a polarizer selectively filters light (see Figure 4a), we design our own "polarizer", i.e., the **Dual-Prism** (DP) method for graph data augmentation, as depicted in Figure 4b. The details are illustrated below, followed by both empirical evidence and theoretical rationale to substantiate our approach.

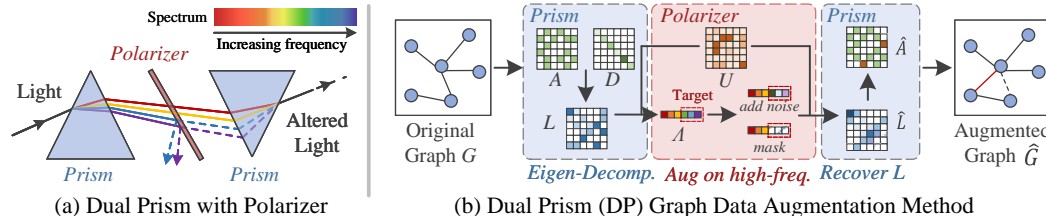

(a) Dual Prism with Polarizer        (b) Dual Prism (DP) Graph Data Augmentation Method

Figure 4: The framework of our Dual-Prism (DP) for graph data augmentation.

### 4.1 PROPOSED AUGMENTATION METHODS

The proposed Dual-Prism (DP) methods, including *DP-Noise* and *DP-Mask*, obtain the augmented graphs by directly changing the spectrum of graphs. A step-by-step breakdown is delineated in Algorithm 1. The DP method starts by extracting the Laplacian Matrix $L$ of a graph $\mathcal{G}$ by $L = D - A$ and then computes its eigen-decomposition $L = U\Lambda U^\top$. Based on the frequency ratio $r_f$, $N_a$ eigenvalues are selected for augmentation, where $N_a = N \times r_f$. Note that since we only target high-frequency eigenvalues, we arrange eigenvalues in increasing order and only focus on the *last* $N_a$ eigenvalues. Then, a binary mask $M$ is formed based on the augmentation ratio $r_a$ to sample the eigenvalues to make a change. Depending on the chosen augmentation type $T$, either noise is infused to the sampled eigenvalues, modulated by $\sigma$ and $M$ (from Line 6 to 8), or the eigenvalues are adjusted using the mask $M$ directly (from Line 9 to 10). Finally, we reconstruct the new Laplacian $\hat{L}$ based on the updated eigenvalues $\hat{\Lambda}$. Given the Laplacian matrix is $L = D - A$, where $D$ is a diagonal matrix, an updated adjacency matrix $\hat{A}$ can be derived by eliminating self-loops. Lastly, we obtain the augmented graph $\hat{\mathcal{G}}$ with its original labels and features retained.

**Selection of $L$.** Note that we adopt the Laplacian matrix $L$ instead of the normalized Laplacian matrix $L_{\text{norm}} = I - D^{-1/2}AD^{-1/2}$. The rationale behind this choice is that reconstructing the adjacency matrix using $L_{\text{norm}}$ necessitates solving a system quadratic equation, where the number of unknown parameters equals the number of nodes in the graph. The computational complexity of this solution is more than $O(N^3)$. Even if we approximate it as a quadratic optimization problem, making it solvable with a gradient-based optimizer, the computational overhead introduced by solving such an optimization problem for each graph to be augmented is prohibitively high.

---

**Algorithm 1** Dual-Prism Augmentation

---

**Input:** Graph $\mathcal{G}$, Frequency Ratio $r_f$, Augmentation Ratio $r_a$, Standard Deviation $\sigma$, Augmentation Type $T$.
1:   $N \leftarrow$ the number of nodes in $\mathcal{G}$
2:   $L \leftarrow$ Laplacian Matrix of $\mathcal{G}$
3:   $U$ and $\Lambda \leftarrow$ eigenvalue decomposition of $L$               $\triangleright$ $\Lambda$ is arranged in increasing order.
4:   $N_a \leftarrow int(N \times r_f)$               $\triangleright$ Get the number of eigenvalues to be augmented.
5:   $M \leftarrow \{m_i \sim Bern(r_a)\}_{i=1}^{N_a}$
6:   **if** $T =$ noise **then**
7:      $\epsilon \leftarrow \{\epsilon_i \sim \mathcal{N}(0,1)\}_{i=1}^{N_a}$
8:      **for** $i \in \{1, \cdots, N_a\}$ **do** $\{\lambda_{N-i} \leftarrow \max(0, \lambda_{N-i} + \sigma M_i \epsilon_i)\}$   $\triangleright$ Add noise to the high-frequency part.
9:   **else if** $T =$ mask **then**
10:      **for** $i \in \{1, \cdots, N_a\}$ **do** $\{\lambda_{N-i} \leftarrow (1 - M_i)\lambda_i\}$         $\triangleright$ Mask the high-frequency part.
11:   $\hat{L} \leftarrow U^\top \hat{\Lambda} U, \hat{A} \leftarrow -\hat{L}$           $\triangleright$ Calculate the new Laplacian and new adjacent matrix.
12:   **for** $i \in \{1, \cdots, N\}$ **do** $\{\hat{A}_{ii} \leftarrow 0\}$
**Output:** Augmented $\hat{\mathcal{G}}$ with edge_index derived from $\hat{A}$, and the label and features unchanged

---

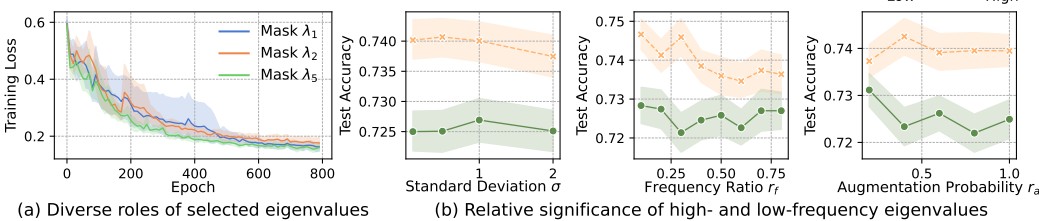

Figure 5: (a) Training loss of GIN model on REDDIT-BINARY when graphs are augmented by masking different eigenvalues. (b) Graph classification performance on IMDB-BINARY under various hyperparameters. The lines represent the average accuracy, while the shaded areas indicates the error margins.

## 4.2 EMPIRICAL EVIDENCE

Next, we aim to verify the correctness of our methods on the improvement of the performance for graph classification via experimental analysis.

**Diverse roles of eigenvalues.** We begin by masking selected eigenvalues to generate graphs for 20% of REDDIT-BINARY. The training loss when masking the eigenvalues $\lambda_1$, $\lambda_2$ and $\lambda_5$ is shown in Figure 5a. These curves suggest that individual eigenvalues contribute differently to the training process. Specifically, masking $\lambda_1$ results in a notably unstable training loss for the initial 500 epochs, evidenced by the expansive blue-shaded region. For $\lambda_2$, while the shaded region's extent is smaller, the curve exhibits noticeable fluctuations, particularly around epoch 200. Conversely, when masking $\lambda_5$, the training appears more stable, with the green curve showing relative steadiness and a reduced shaded area. These demonstrate the various significance of eigenvalues: $\lambda_1$ and $\lambda_2$ include more crucial structural and property details of the graph compared to $\lambda_5$. As a result, they deserve to be prioritized for preservation during augmentation.

**Different importance of high- and low-frequency parts.** We then conduct experiments on group-level eigenvalues, i.e., the high- and low-frequency eigenvalues, to gain a broader view of the influence exerted by varying frequency eigenvalues. We introduce noise to eigenvalues across various hyperparameter combinations. Concretely, we use the standard deviation $\sigma$ to determine the magnitude of the noise. The frequency ratio $r_f$ dictates the number of eigenvalues $N_a$ we might change, while the augmentation probability $r_a$ specifies the final eigenvalues sampled for modification. The eigenvalues are arranged in ascending order. The setting of 'Low' means that we select candidates from the first $N_a$ eigenvalues, while 'High' denotes selection from the last $N_a$ eigenvalues. As shown in Figure 5b, the orange lines consistently outperform the green lines across all three plots, indicating that the performance associated with perturbing high-frequency eigenvalues consistently exceeds that of their low-frequency counterparts. Moreover, as the frequency ratio $r_f$ increases, the accuracy of the 'Low' scenario remains relatively stable and low. Contrastingly, for the 'High' scenario, a notable decline in accuracy begins once the ratio exceeds around 30%. This suggests that the eigenvalues outside the top 30% of the high-frequency range may start to include more critical information beneficial for graph classification tasks that should not be distorted.

## 4.3 THEORETICAL BACKING AND INSIGHTS

The eigenvalues of the Laplacian matrix provide significant insights into various graph properties, as established in prior research (Chung, 1997). Such insights have driven and backed our proposal to modify the high-frequency eigenvalues while preserving their low-frequency counterparts. For example, the second-smallest eigenvalue $\lambda_1$, often termed the *Fiedler value*, quantifies the graph's *algebraic connectivity*. A greater Fiedler value indicates a better-connected graph and it is greater than 0 if and only if the graph is a connected graph. The number of times 0 appears as an eigenvalue is the number of connected components in the graph. In addition to the connectivity, the diameter of a graph is also highly related to the eigenvalues – it can be upper and lower bounded from its spectrum (Chung, 1997): $4/n\lambda_1 \leq d \leq 2[\sqrt{2m/\lambda_1}\log_2 n]$, where $n$ and $m$ denotes the number of nodes and the maximum degree of the graph, respectively. In addition to these widely-used properties, other properties are also highly related to spectrum, including graph diffusion distance (Hammond et al., 2013). In essence, eigenvalues serve as powerful spectral signatures comprising a myriad of structural and functional aspects of graphs.

## 5 EXPERIMENTS

**Experimental Setup.** We evaluate our augmentation method for graph classification tasks under four different settings, including supervised learning, semi-supervised learning, unsupervised learning, and transfer learning. We conduct our experiments on 21 real-world datasets across three different domains, including bio-informatics, molecule, and social network, from the TUDatasets benchmark (Morris et al., 2020), OGB benchmark (Hu et al., 2020a) and ZINC chemical molecule dataset (Hu et al., 2020b). The details of the datasets, baselines, and experimental settings can be found in Appendices A, B, and C, respectively. More empirical results, including the evaluations on hyperparameter sensitivity analysis, can be found in Appendix D.

### 5.1 SUPERVISED LEARNING

**Performance.** We first evaluate our proposed methods in the supervised learning setting. Following the prior works (Han et al., 2022; Ling et al., 2023), we use GIN and GCN as backbones for graph classification on eight different datasets. Table 1 shows the performance of our proposed methods compared with seven state-of-art (SOTA) baselines, including DropEdge(Rong et al., 2019), DropNode(Feng et al., 2020), Subgraph(You et al., 2020), M-Mixup (Verma et al., 2019), Sub-Mix(Yoo et al., 2022), G-Mixup(Han et al., 2022) and S-Mixup (Ling et al., 2023). According to the results, our DP-Noise method consistently outperforms other existing methods across the majority of datasets, establishing its dominance in effectiveness. DP-Mask also shines, often securing a noteworthy second-place standing. GIN tends to obtain superior outcomes, especially when combined with DP-Noise, which is exemplified by its 61.67% classification accuracy on IMDB-M. Note that on REDD-B, GIN achieves more satisfactory performance than GCN, which is a consistent pattern across baselines but becomes particularly pronounced with our methods. This phenomenon may be attributed to the intrinsic characteristics of GIN and GCN. GIN is known for its precision in delineating complex structural intricacies of graphs (Xu et al., 2018), while GCN is characterized by its smoothing effect (Defferrard et al., 2016). Our methods' superiority in diversifying the graphs' structures naturally amplifies GIN's strengths. In contrast, GCN may not be as adept at leveraging the enhancements offered by our augmentation techniques.

**Generalization.** Figures 6a and 6b display the test loss and accuracy curves for the IMDB-B dataset, comparing four distinct augmentation strategies: G-mixup, DP-Noise, DP-Mask, and a scenario without any augmentation (i.e., Vanilla). A notable trend is the consistently lower and more stable test loss curves for DP-Noise and DP-Mask in comparison to Vanilla and G-mixup. Concurrently, the accuracy achieved with DP-Noise and DP-Mask is higher. This indicates our proposed methods' superior generalization and the capacity for enhancing model stability.

### 5.2 SEMI-SUPERVISED LEARNING

**Performance.** We then evaluate our methods in a semi-supervised setting comparing with five baselines, including training from scratch without and with augmentations (denoted as Vanilla and Aug., respectively), GAE (Kipf & Welling, 2016b), Informax (Veličković et al., 2018) and GraphCL(You

Table 1: Performance comparisons with GCN and GIN in the *supervised* learning setting. The best and second best results are highlighted with **bold** and underline, respectively. * and ** denote the improvement over the second best baseline is statistically significant at level 0.1 and 0.05, respectively(Newey & West, 1987). Baseline results are taken from Ling et al. (2023); Han et al. (2022).

| | Dataset | IMDB-B | IMDB-M | REDD-B | REDD-M5 | REDD-M12 | PROTEINS | NCI1 | ogbg-molhiv |
|---|---|---|---|---|---|---|---|---|---|
| GCN | Vanilla | 72.80±4.08 | 49.47±2.60 | 84.85±2.42 | 49.99±1.37 | 46.90±0.73 | 71.43±2.60 | 72.38±1.45 | - |
| | DropEdge | 73.20±5.62 | 49.00±2.94 | 85.15±2.81 | 51.19±1.74 | 47.08±0.55 | 71.61±4.28 | 68.32±1.60 | - |
| | DropNode | 73.80±5.71 | 50.00±4.85 | 83.65±3.63 | 47.71±1.75 | 47.93±0.64 | 72.69±3.55 | 70.73±2.02 | - |
| | Subgraph | 70.90±5.07 | 49.80±3.43 | 68.41±2.57 | 47.31±5.23 | 47.49±0.93 | 67.93±3.24 | 65.05±4.36 | - |
| | M-Mixup | 72.00±5.66 | 49.73±2.67 | 87.05±2.47 | 51.49±2.00 | 46.92±1.05 | 71.16±2.87 | 71.58±1.79 | - |
| | SubMix | 72.30±4.75 | 49.73±2.88 | 85.15±2.37 | 52.87±2.19 | - | 72.42±2.43 | 71.65±1.58 | - |
| | G-Mixup | 73.20±5.60 | 50.33±3.67 | 86.85±2.30 | 51.77±1.42 | 48.06±0.53 | 70.18±2.44 | 70.75±1.72 | - |
| | S-Mixup | 74.40±5.44 | 50.73±3.66 | **89.30±2.69** | 53.29±1.97 | - | 73.05±2.81 | **75.47±1.49** | 96.70±0.20 |
| | DP-Noise | **77.90±2.30** * | **53.60±1.59** ** | 84.60±7.61 | **53.42±1.36** | **48.47±0.57** | **75.03±2.66** | 69.20±2.57 | **97.02±0.19** ** |
| | DP-Mask | 76.00±3.62 | 51.20±1.73 | 76.70±1.34 | 52.42±2.78 | 47.25±1.12 | 73.60±3.10 | 62.45±3.80 | 96.90±0.24* |
| GIN | Vanilla | 71.30±4.36 | 48.80±2.54 | 89.15±2.47 | 53.17±2.26 | 50.23±0.83 | 68.28±2.47 | 79.08±2.12 | - |
| | DropEdge | 70.50±3.80 | 48.73±4.08 | 87.45±3.91 | 54.11±1.94 | 49.77±0.76 | 68.01±3.22 | 76.47±2.34 | - |
| | DropNode | 72.00±6.97 | 45.67±2.59 | 88.60±2.52 | 53.97±2.11 | 49.95±1.70 | 69.64±2.98 | 74.60±2.12 | - |
| | Subgraph | 70.40±4.98 | 43.74±5.74 | 76.80±3.87 | 50.09±4.94 | 49.67±0.90 | 66.67±3.10 | 60.17±2.33 | - |
| | M-Mixup | 72.00±5.14 | 48.67±5.32 | 87.70±2.50 | 52.85±1.03 | 49.81±0.80 | 68.65±3.76 | 79.85±1.88 | - |
| | SubMix | 71.70±6.20 | 49.80±4.01 | 90.45±1.93 | 54.27±2.92 | - | 69.54±3.15 | 79.78±1.09 | - |
| | G-Mixup | 72.40±5.64 | 49.93±2.82 | 90.20±2.84 | 54.33±1.99 | 50.50±0.41 | 64.69±3.60 | 78.20±1.58 | - |
| | S-Mixup | 73.40±6.26 | 50.13±4.34 | 90.55±2.11 | 55.19±1.99 | - | 69.37±2.86 | 80.02±2.45 | 96.84±0.40 |
| | DP-Noise | **78.40±1.82** ** | **61.67±0.71** ** | **93.42±1.41** ** | **57.72±1.87** ** | **53.70±1.16** ** | **73.51±4.54** ** | **90.56±5.78** ** | **97.43±0.48** ** |
| | DP-Mask | 76.30±2.56 | 51.60±1.32 | 93.25±1.19** | 56.50±0.80* | 49.11±1.30 | 72.79±1.95** | 80.30±2.01 | 96.98±0.29 |

Figure 6: (a) The loss and (b) accuracy curves for supervised learning on test data of IMDB-BINARY using GIN, with four augmentation methods. Curves represent mean values from 5 runs, and shaded areas indicate standard deviation. The accuracy gain (%) in semi-supervised learning when contrasting 5 different augmentation methods with (c) DP-Noise and (b) DP-Mask across 7 datasets. Warmer color means better performance gains.

et al., 2020). Table 2 provides the performance comparison when utilizing 1% and 10% label ratios. At the more challenging 1% label ratio, our DP-Noise achieves SOTA results across all three datasets, and DP-Mask secures the second-best performance on two out of the three datasets. As the label ratio increases to 10%, DP-Noise maintains its efficacy, showcasing excellent performance on six out of seven datasets.

**Augmentation Pairing Efficiency.** To investigate optimal combinations that could potentially enhance performance, we evaluate the synergistic effects on accuracy gain (%) when pairing DP-Noise and DP-Mask with different augmentation methods using the same setting in You et al. (2020). From Figures 6c and 6d, overall, the diverse accuracy gains across datasets indicate that there is no one-size-fits-all "partner" for DP-Noise and DP-Mask; the efficacy of each combination varies depending on the dataset. However, generally, both DP-Noise and DP-Mask exhibit enhanced performance when paired with dropN at a large degree.

## 5.3 UNSUPERVISED REPRESENTATION LEARNING

We next evaluate our strategies in the unsupervised learning setting and compare them with 12 baselines, including three graph kernel methods (GL (Pržulj, 2007), WL(Shervashidze et al., 2011) and DGK(Yanardag & Vishwanathan, 2015)), four representation learning methods (node2vec (Grover & Leskovec, 2016), sub2vec(Adhikari et al., 2018), graph2vec(Narayanan et al., 2017) and InfoGraph(Sun et al., 2019)) and five GCL-based methods (GraphCL(You et al., 2020), MV-GRL(Hassani & Khasahmadi, 2020), AD-GCL(Suresh et al., 2021), JOAO(You et al., 2021) and GCL-SPAN(Lin et al., 2022)). Table 3 shows the performance of our methods in an unsupervised setting. From the results, DP-Noise and DP-Mask surpass other baselines on five out of seven datasets. Notably, compared with another spectral-based method GCL-SPAN (Lin et al., 2022), our methods outperform it on most datasets, especially on the molecules dataset NCI1 (increase of around 11.5% accuracy). This can be explained by that despite GCL-SPAN is also spectral-based, it in fact modifies the spatial graph while optimizing in the spectral realm. In contrast, our methods directly make alterations in the spectral domain to preserve the structural information. Given the

Table 2: Performance comparisons in the *semi-supervised* learning setting. The best and second best results are highlighted with **bold** and underline, respectively. 1% or 10% is the label ratio. The metric is accuracy (%). * and ** denote the improvement over the second best baseline is statistically significant at level 0.1 and 0.05, respectively. Baseline results are taken from You et al. (2020).

| Dataset | NCI1 | PROTEINS | DD | COLLAB | REDD-B | REDD-M5 | GITHUB |
|---|---|---|---|---|---|---|---|
| 1% Vallina | 60.72±0.45 | - | - | 57.46±0.25 | - | - | 54.25±0.22 |
| 1% Aug. | 60.49±0.46 | - | - | 58.40±0.97 | - | - | 56.36±0.42 |
| 1% GAE | 61.63±0.84 | - | - | 63.20±0.67 | - | - | 59.44±0.44 |
| 1% Infomax | 62.72±0.65 | - | - | 61.70±0.77 | - | - | 58.99±0.50 |
| 1% GraphCL | 62.55±0.86 | - | - | 64.57±1.15 | - | - | 58.56±0.59 |
| 1% DP-Noise | **63.43**±1.39 | **-** | **-** | **65.94**±3.13 | **-** | **-** | **60.06**±2.72 |
| 1% DP-Mask | 62.43±1.08 | **-** | **-** | 65.68±1.66 * | **-** | - | 59.70±0.53 |
| 10% Vallina | 73.72±0.24 | 70.40±1.54 | 73.56±0.41 | 73.71±0.27 | 86.63±0.27 | 51.33±0.44 | 60.87±0.17 |
| 10% Aug. | 73.59±0.32 | 70.29±0.64 | 74.30±0.81 | 74.19±0.13 | 87.74±0.39 | 52.01±0.20 | 60.91±0.32 |
| 10% GAE | 74.36±0.24 | 70.51±0.17 | 74.54±0.68 | 75.09±0.19 | 87.69±0.40 | 53.58±0.13 | 63.89±0.52 |
| 10% Infomax | 74.86±0.26 | 72.27±0.40 | 75.78±0.34 | 73.76±0.29 | 88.66±0.95 | 53.61±0.31 | 65.21±0.88 |
| 10% GraphCL | 74.63±0.25 | 74.17±0.34 | 76.17±1.37 | 74.23±0.21 | 89.11±0.19 | 52.55±0.45 | **65.81**±0.79 |
| 10% DP-Noise | **75.30**±0.58** | **74.73**±1.01 | **76.91**±0.81 | **77.05**±0.82** | **89.38**±0.95 | **54.45**±0.64** | 65.59±0.88 |
| 10% DP-Mask | 74.88±1.84 | 71.37±4.18 | 75.64±0.81 | 76.90±0.62** | 88.62±0.63 | 52.80±0.59 | 64.95±1.03 |

critical role that structural patterns in molecular data play on classification tasks, the enhanced performance underscores the efficiency of our direct spectral modifications in creating more effective and insightful augmented graphs for graph classification.

Table 3: Performance comparisons in the *unsupervised* learning results. The best and second best results are highlighted with **bold** and underline, respectively. The metric is accuracy (%). * and ** denote the improvement over the second best baseline is statistically significant at level 0.1 and 0.05, respectively. Baseline results are taken from You et al. (2020); Lin et al. (2022).

| Dataset | NCI1 | PROTEINS | DD | MUTAG | REDD-B | REDD-M5 | IMDB-B |
|---|---|---|---|---|---|---|---|
| GL | - | - | - | 81.66±2.11 | 77.34±0.18 | 41.01±0.17 | 65.87±0.98 |
| WL | 80.01±0.50 | 72.92±0.56 | - | 80.72±3.00 | 68.82±0.41 | 46.06±0.21 | 72.30±3.44 |
| DGK | **80.31±0.46** | 73.30±0.82 | - | 87.44±2.72 | 78.04±0.39 | 41.27±0.18 | 66.96±0.56 |
| node2vec | 54.89±1.61 | 57.49±3.57 | - | 72.63±10.20 | - | - | - |
| sub2vec | 52.84±1.47 | 53.03±5.55 | - | 61.05±15.80 | 71.48±0.41 | 36.68±0.42 | 55.26±1.54 |
| graph2vec | 73.22±1.81 | 73.30±2.05 | - | 83.15±9.25 | 75.78±1.03 | 47.86±0.26 | 71.10±0.54 |
| InfoGraph | 76.20±1.06 | 74.44±0.31 | 75.23±0.39 | 89.01±1.13 | 82.50±1.42 | 53.46±1.03 | 73.03±0.87 |
| GraphCL | 77.87±0.41 | 74.39±0.45 | 78.62±0.40 | 86.80±1.34 | 89.53±0.84 | **55.99±0.28** | 71.14±0.44 |
| MVGRL | 68.68±0.42 | 74.02±0.32 | 75.20±0.55 | 89.24±1.31 | 81.20±0.69 | 51.87±0.65 | 71.84±0.78 |
| AD-GCL | 69.67±0.51 | 73.59±0.65 | 74.49±0.52 | 89.25±1.45 | 85.52±0.79 | 53.00±0.82 | 71.57±1.01 |
| JOAO | 72.99±0.75 | 71.25±0.85 | 66.91±1.75 | 85.20±1.64 | 78.35±1.38 | 45.57±2.86 | 71.60±0.86 |
| GCL-SPAN | 71.43±0.49 | **75.78±0.41** | 75.78±0.52 | 89.12±0.76 | 83.62±0.64 | 54.10±0.49 | **73.65±0.69** |
| DP-Noise | 79.69±0.70 | 74.60±0.43 | 78.59±0.23 | 87.63±1.98 | 90.90±0.32** | 55.54±0.15 | 71.42±0.41 |
| DP-Mask | 79.47±0.22 | 74.70±0.29 | **79.97±1.09** | **89.98±1.36** | **91.21±0.24** | 55.92±0.49 | 71.78±0.37 |

## 5.4 TRANSFER LEARNING

We lastly conduct transfer learning experiments on molecular property prediction in the manner of Hu et al. (2020b) to evaluate the capability of our methods for learning generalizable encoders. Specifically, we initially pre-train models on the extensive chemical molecule dataset ZINC (Sterling & Irwin, 2015), then fine-tune the models on eight distinct datasets within a similar domain. We draw comparisons between our methods and six baselines, including a reference model without pre-training (referred to *No-Pre-Train*), Informax (Veličković et al., 2018), EdgePred (Hamilton et al., 2017), AttrMasking (Hu et al., 2020b), ContextPred (Hu et al., 2020b) and GraphCL You et al. (2020). The comparative results are shown in Table 4. Our methods demonstrated SOTA performance, outperforming competitors on half of the datasets. Especially, our methods consistently outperform the conventional GraphCL method, which indicates our data augmentation methods as better choices for graph contrastive learning. Notably, DP-Mask achieves an 83.52% ROC-AUC score on ClinTox, exceeding the performance of GraphCL by a substantial margin (nearly 10%). These findings demonstrate the enhanced efficacy of our techniques in the transfer learning setting for graph classification tasks.

Table 4: Performance comparisons in the *transfer* learning setting. The best and second best results are highlighted with **bold** and underline, respectively. The metric is ROC-AUC scores (%). * and ** denote the improvement over the second best baseline is statistically significant at level 0.1 and 0.05, respectively. Baseline results are taken from Hu et al. (2020b); You et al. (2020).

| Dataset | BBBP | Tox21 | ToxCast | SIDER | ClinTox | MUV | HIV | BACE |
|---|---|---|---|---|---|---|---|---|
| No-Pre-Train | 65.80±4.50 | 74.00±0.80 | 63.40±0.60 | 57.30±1.60 | 58.00±4.40 | 71.80±2.50 | 75.30±1.90 | 70.10±5.40 |
| Informax | 68.80±0.80 | 75.30±0.50 | 62.70±0.40 | 58.40±0.80 | 69.90±3.00 | 75.30±2.50 | 76.00±0.70 | 75.90±1.60 |
| EdgePred | 67.30±2.40 | 76.00±0.60 | 64.10±0.60 | 60.40±0.70 | 64.10±3.70 | 74.10±2.10 | 76.30±1.00 | **79.90±0.90** |
| AttrMasking | 64.30±2.80 | **76.70±0.40** | **64.20±0.50** | 61.00±0.70 | 71.80±4.10 | 74.70±1.40 | 77.20±1.10 | 79.30±1.60 |
| ContextPred | 68.00±2.00 | 75.70±0.70 | 63.90±0.60 | 60.90±0.60 | 65.90±3.80 | **75.80±1.70** | 77.30±1.00 | 79.60±1.20 |
| GraphCL | 69.68±0.67 | 73.87±0.66 | 62.40±0.57 | 60.53±0.88 | 75.99±2.65 | 69.80±2.66 | 78.47±1.22 | 75.38±1.44 |
| DP-Noise | 70.38±0.91* | 74.33±0.42 | 64.08±0.25 | 61.52±0.79 | 76.26±1.68 | 73.39±2.08 | 78.63±0.37 | 76.23±0.86 |
| DP-Mask | **71.63±1.86** | 74.91±0.49 | 63.43±0.28 | 61.33±0.17 | **83.52±1.07** | 73.77±1.40 | 77.80±1.31 | 78.73±1.13 |

## 6 CONCLUSION

In this study, we adopt a spectral perspective, bridging graph properties and spectral insights for property-retentive and globally-aware graph data augmentation. Stemming from this point, we propose a novel augmentation method called Dual-Prism (DP), including DP-Noise and DP-Mask. By focusing on different frequency components in the spectrum, our method skillfully preserves graph properties while ensuring diversity in augmented graphs. Our extensive evaluations validate the efficacy of our methods across various learning paradigms on the graph classification task. In summary, our contributions highlight the potential of leveraging spectral insights in graph data augmentation.

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

## A    DETAILS OF DATASETS

We conduct our experiments on 21 different graph real-world datasets for graph classification tasks. In this section, we provide detailed descriptions of the datasets used in this paper. Specifically, for the *supervised* learning setting, we include a total of eight datasets from the TUDatasets benchmark (Morris et al., 2020) (i.e., PROTEINS, NCI1, IMDB-BINARY, IMDB-MULTI, REDDIT-BINARY, REDDIT-MULTI-5K, and REDDIT-MULTI-12K) and the OGB benchmark (Hu et al., 2020a) (i.e., ogbg-molhiv). For the *semi-supervised* learning setting, we include seven different datasets from the TUDatasets benchmark (Morris et al., 2020) (i.e., PROTEINS, NCI1, DD, COLLAB, GITHUB, REDDIT-BINARY, and REDDIT-MULTI-5K). For the *unsupervised* learning setting, we include seven different datasets from the TUDatasets benchmark (Morris et al., 2020) (i.e., PROTEINS, NCI1, DD, MUTAG, IMDB-BINARY, REDDIT-BINARY, and REDDIT-MULTI-5K). The detailed of these datasets can be found in Table 5.

Table 5: Statistical characteristics of the datasets in three learning settings. Supe.: Supervised learning. Semi.: Semi-supervised learning. Unsu.: Unsupervised learning.

| Dataset | Category | # Graphs | # Avg edges | # Classes | Task Supe. | Semi. | Unsu. |
|---------|----------|----------|-------------|-----------|------|-------|-------|
| IMDB-BINARY | Social Networks | 1,000 | 96.53 | 2 | ✓ | | ✓ |
| IMDB-MULTI | Social Networks | 1,500 | 65.94 | 3 | ✓ | | |
| REDDIT-BINARY | Social Networks | 2,000 | 497.75 | 2 | ✓ | ✓ | ✓ |
| REDDIT-MULTI-5K | Social Networks | 4,999 | 594.87 | 5 | ✓ | ✓ | ✓ |
| REDDIT-MULTI-12K | Social Networks | 11,929 | 456.89 | 11 | ✓ | | |
| COLLAB | Social Networks | 5,000 | 2457.78 | 3 | | ✓ | |
| GITHUB | Social Networks | 12,725 | 234.64 | 2 | | ✓ | |
| DD | Biochemical Molecules | 1,178 | 715.66 | 2 | | ✓ | ✓ |
| MUTAG | Biochemical Molecules | 188 | 19.79 | 2 | | | ✓ |
| PROTEINS | Biochemical Molecules | 1,113 | 72.82 | 2 | ✓ | ✓ | ✓ |
| NCI1 | Biochemical Molecules | 4,110 | 32.30 | 2 | ✓ | ✓ | ✓ |
| ogbg-molhiv | Biochemical Molecules | 41,127 | 27.50 | 2 | ✓ | | |

For the *transfer* learning setting, we pre-train on ZINC-2M chemical molecule dataset (Sterling & Irwin, 2015; Gómez-Bombarelli et al., 2018), and fine-tune on eignt different datasets, namely BBBP, Tox21, ToxCast, SIDER, ClinTox, MUV, HIV, and BACE. The detailed of these datasets can be found in Table 6.

Table 6: Statistical characteristics of the datasets used in the transfer learning setting.

| Dataset | Strategy | # Molecules | # Binary tasks |
|---------|----------|-------------|----------------|
| ZINC-2M | Pre-training | 2,000,000 | - |
| BBBP | Fine-tuning | 2,039 | 1 |
| Tox21 | Fine-tuning | 7,831 | 12 |
| ToxCast | Fine-tuning | 8,576 | 617 |
| SIDER | Fine-tuning | 1,427 | 27 |
| ClinTox | Fine-tuning | 1,477 | 2 |
| MUV | Fine-tuning | 93,087 | 17 |
| HIV | Fine-tuning | 41,127 | 1 |
| BACE | Fine-tuning | 1,513 | 1 |

## B    DETAILS OF BASELINES

**Supervised learning.** For experiments in the supervised setting, we select the following baseline:

* DropEdge (Rong et al., 2019) selectively drops a portion of edges from the input graphs.

* DropNode (Feng et al., 2020) omits a specific ratio of nodes from the provided graphs.

* Subgraph (You et al., 2020) procures subgraphs from the main graphs using a random walk sampling technique.

* M-Mixup (Verma et al., 2019) blends graph-level representations through linear interpolation.

* SubMix (Yoo et al., 2022) combines random subgraphs from paired input graphs.

- G-Mixup (Han et al., 2022) employs a class-focused graph mixup strategy by amalgamating graphons across various classes.
- S-Mixup (Ling et al., 2023) adopts a mixup approach for graph classification, emphasizing soft alignments.

**Semi-supervised learning.** For experiments in the semi-supervised setting, we select the following baseline methods:

- GAE (Kipf & Welling, 2016b) is a non-probabilistic graph auto-encoder model, which is a variant of the VGAE (variational graph autoencoder).
- Informax (Veličković et al., 2018) trains a node encoder to optimize the mutual information between individual node representations and a comprehensive global graph representation.
- GraphCL (You et al., 2020) conducts an in-depth exploration of graph structure augmentations, including random edge removal, node dropping, and subgraph sampling.

**Unsupervised learning.** For experiments in the unsupervised setting, we select the following baseline methods:

- GL (graphlet kernel) (Pržulj, 2007) measures the similarity between graphs by counting the occurrences of small subgraphs, known as graphlets, within them. It captures local topological patterns, thus providing a comprehensive view of the graph structure.
- WL (Weisfeiler-Lehman sub-tree kernel) (Shervashidze et al., 2011) captures the similarity between graphs by comparing subtrees of increasing heights. It effectively distinguishes non-isomorphic graphs and is often employed for graph classification tasks.
- DGK (deep graph kernel) (Yanardag & Vishwanathan, 2015) combines the strengths of both graph kernels and deep learning. It leverages convolutional neural networks to learn hierarchical representations of graphs, enabling the kernel to capture complex patterns and structures within the data for a more refined similarity measure.
- node2vec (Grover & Leskovec, 2016) captures low-dimensional embeddings of graph nodes by leveraging random walks originating from target nodes.
- sub2vec (Adhikari et al., 2018) seeks to capture feature representations of arbitrary subgraphs, addressing the limitations of node-centric embeddings.
- graph2vec (Narayanan et al., 2017) is a neural embedding framework designed to learn data-driven distributed representations of entire graphs.
- InfoGraph (Sun et al., 2019) is designed to maximize the mutual information between complete graph representations and various substructures, such as nodes, edges, and triangles.
- GraphCL (see above section).
- MVGRL (Hassani & Khasahmadi, 2020) establishes a link between the local Laplacian matrix and a broader diffusion matrix by leveraging mutual information. This approach yields representations at both the node and graph levels, catering to distinct prediction tasks.
- AD-GCL (Suresh et al., 2021) emphasize preventing the capture of redundant information during training. They achieve this by optimizing adversarial graph augmentation strategies in GCL and introducing a trainable non-i.i.d. edge-dropping graph augmentation.
- JOAO (You et al., 2021) utilize a bi-level optimization framework to sift through optimal strategies, exploring multiple augmentation types like uniform edge or node dropping and subgraph sampling.
- GCL-SPAN (Lin et al., 2022) introduces a spectral augmentation approach, which directs topology augmentations to maximize spectral shifts.

**Transfer learning.** For experiments in the transfer setting, we select the following baseline methods:

- Informax (see above section).

- EdgePred (Hamilton et al., 2017) employs an inductive approach that utilizes node features, such as text attributes, to produce node embeddings by aggregating features from a node's local neighborhood, rather than training distinct embeddings for each node.

- AttrMasking (Attribute Masking) (Hu et al., 2020b) is a pre-training method for GNNs that harnesses domain knowledge by discerning patterns in node or edge attributes across graph structures.

- ContextPred (Context Prediction) (Hu et al., 2020b) is designed for pre-training GNNs that simultaneously learn local node-level and global graph-level representations via subgraphs to predict their surrounding graph structures.

- GraphCL (see above section).

## C  DETAILS OF EXPERIMENTS SETTINGS

We conduct our experiments with PyTorch 1.13.1 on a server with NVIDIA RTX A5000 and CUDA 12.2. For each experiment, we run 10 times. We detail the settings of our experiments in this paper as follows.

**Speed up implementation.** Our augmentation method involves matrix eigenvalue decomposition, which is highly CPU-intensive. During implementation, we observed that when there is insufficient CPU, parallelly executing numerous matrix eigenvalue decomposition can lead to CPU resource deadlock. To address this issue, we established an additional set of CPU locks to manage CPU scheduling. Let $N_{cpu}$ represent the number of CPUs available for each matrix eigenvalue decomposition task, and $N_{parallel}$ denote the number of decomposition tasks that can be executed simultaneously. We set $N_{cpu} \times N_{parallel}$ to be less than the total CPU number of the server. During task execution, we created a list of CPU locks, with each lock corresponding to $N_{cpu}$ available CPUs. There is no overlap between the CPUs corresponding to each lock. Before a matrix decomposition task is executed, it must first request a CPU lock. Once the lock is acquired, the task can only be executed on the designated CPUs. After completion, the task releases the CPU lock. If there are no free locks in the current CPU lock list, the matrix decomposition task must wait. By employing this approach, we effectively isolated parallel matrix decomposition tasks.

**Empirical studies.** We first detail the processes and settings of the empirical studies below.

- **Experiment of Figure 1.** For DropEdge, 20% edges are randomly dropped. For DP-Noise, we use a standard deviation of 7, an augmentation probability of 0.5, and an augmentation frequency ratio of 0.5. For DP-Mask, the augmentation probability is set to 0.3, with an augmentation frequency ratio of 0.4. Detailed variations in edge numbers and properties between the original and augmented graphs can be found in Table 7.

- **Experiment of Figure 3.** In Figures 3a and 3b, labels in REDDIT-MULTI-12K for Class A and Class B are 1 and 10, respectively. The added edges in 3c are 3↔5, 3↔7, 1↔6, 2↔6, 0↔2, 1↔3, 1↔4, 2↔4, 4↔6, and 5↔7, respectively. The dropped edges in 3d are 0↔4, 3↔4, 4↔5, 4↔7, 0↔1, 2↔3, 0↔3, 5↔6, 6↔7, and 1↔2, respectively. The change of spectrum $\Delta L_2$ is the $L_2$ distance between the spectrum of $\mathcal{G}$ and augmented graph $\mathcal{G}'$, denoted as $\Delta L_2 = \sqrt{\sum_i (\lambda_i(\mathcal{G}) - \lambda_i(\hat{\mathcal{G}}))^2}$, where $\lambda(\mathcal{G})$ represent the spectrum of $\mathcal{G}$ and $\lambda(\hat{\mathcal{G}})$ is the spectrum of $\hat{\mathcal{G}}$.

- **Experiment of Figure 5.** For both Figure 5a and 5b, we employ GIN as the backbone model and conduct experiments over five runs. In Figure 5b, while testing one parameter, we draw the other parameters from their respective search spaces: $\sigma \in [0.1, 0.5, 1.0, 2.0]$, $r_f \in [0.1, 0.2, 0.3, 0.4, 0.5, 0.6, 0.7, 0.8]$, and $r_a \in [0, 0.2, 0.4, 0.6, 0.8, 1]$. In addition, to control the noise adding to low- and high-frequency eigenvalues are in the same scale, the augmented $i$-th eigenvalue is calculated as $\lambda_i = \max(0, 1 + \epsilon) \times \lambda_i$, where $\epsilon \sim \mathcal{N}(0, \sigma)$.

**Supervised learning.** Following prior works (Han et al., 2022; Ling et al., 2023), we utilize two GNN models, namely GCN (Kipf & Welling, 2017) and GIN (Xu et al., 2018). Details of these GNNs are provided below.

- **GCN.** For the TUDatasets benchmark, the backbone model has four GCN layers, utilizes a global mean pooling readout function, has a hidden size of 32, and uses the ReLU activation function. For the ogbg-molhiv dataset, the model consists of five GCN layers, a hidden size of 300, the ReLU activation function, and a global mean pooling readout function.

Table 7: Details of alterations of the number of edges and properties of graphs in Figure 1.

| Graph | Edge Alterations | | | Properties | | | |
|---|---|---|---|---|---|---|---|
| | # Dropped | # Added | Connectivity | Diameter | Radius | # Periphery | ASPL |
| **Original** | - | - | TRUE | 2 | 1 | 11 | 1.42 |
| **DropEdge** | 7 | 0 | TRUE | 3 | 2 | 8 | 1.64 |
| **DP-Noise** | 6 | 0 | TRUE | 2 | 1 | 11 | 1.52 |
| **DP-Mask** | 14 | 4 | TRUE | 2 | 1 | 11 | 1.58 |

- **GIN.** For the TUDatasets benchmark, the backbone model comprises four GIN layers, each with a two-layer MLP. It utilizes a global mean pooling readout function, has a hidden size of 32, and adopts the ReLU activation function. Conversely, for the ogbg-molhiv dataset, the model consists of five GIN layers, a hidden size of 300, the ReLU activation function, and employs a global mean pooling for the readout function.

For all other hyper-parameter search space and training configurations of the experiments on the IMDB-B, IMDB-M, REDD-B, REDD-M5, and REDD-M12, we keep consistent with Han et al. (2022). For all other hyper-parameter search space and training configurations of the experiments on the PROTEIN, NCI1, and ogbg-hiv, we keep consistent with Ling et al. (2023). Note that instead of adopting the results of baseline methods on the ogbg-hiv dataset from the reference directly, we reported the results of rerunning the baseline experiments on the ogbg-hiv dataset, which is higher than the results in the reference.

**Semi-supervised learning.** We maintain consistency with You et al. (2020) for all hyper-parameter search spaces and training configurations. For all datasets, we conduct experiments at label rates of 1% (provided there are more than 10 samples for each class) and 10%. These experiments are performed five times, with each instance corresponding to a 10-fold evaluation. We report both the mean and standard deviation of the accuracies in percentages. During pre-training, we perform a grid search, tuning the learning rate among 0.01, 0.001, 0.0001 and the epoch number within 20, 40, 60, 80, 100.

**Unsupervised representation learning.** Following You et al. (2020); Lin et al. (2022), we use a 5-layer GIN as encoders. For all hyper-parameter search spaces and training settings in unsupervised learning, we also align with the configurations presented in You et al. (2020). We conduct experiments five times, with each iteration corresponding to a 10-fold evaluation. The reported results in Table 3 include both the mean and standard deviation of the accuracy percentages.

**Transfer learning.** Following the transfer learning setting in Hu et al. (2020b); You et al. (2020); Lin et al. (2022), we conduct graph classification experiments on a set of biological and chemical datasets via GIN models. Specifically, an encoder was first pre-trained on the large ZINC-2M chemical molecule dataset (Sterling & Irwin, 2015; Gómez-Bombarelli et al., 2018) and then was evaluated on small datasets from the same domains (i.e., BBBP, Tox21, ToxCast, SIDER, ClinTox, MUV, HIV, and BACE).

## D    MORE EXPERIMENTS RESULTS

**Empirical studies.** In Section 3, we investigate spectral alterations due to adding an edge in a toy graph (depicted in Figure 2a). The spectral changes are presented in Figure 2b. Further, in Figure 7, we analyze the consequences on the spectrum when edges from the same toy graph are removed. Notably, the removal of edges 0-4 and 3-4 results in minimal spectral variations. However, the removal of edges 5-6 and 6-7 leads to pronounced changes in both high and low frequencies, evident from the pronounced shifts in values $\lambda_2$ and $\lambda_5$. To further illustrate our observation, we present an additional example featuring a nine-node toy graph that also exhibits similar results, as shown in Figure 8.

**Hyperparameter sensitivities.** We conducted experiments on the IMDB-BINARY dataset, leveraging various combinations of standard deviation $\sigma$ and frequency ratio $r_f$ for both low and high-frequency components to assess the impacts of DP-Noise parameters. For these experiments, we employed the GIN as our backbone model let $\sigma$ and $r_f$ from two search spaces, where $\sigma \in [0.1, 0.5, 1.0, 2.0]$ and $r_f \in [0.1, 0.2, 0.3, 0.4, 0.5, 0.6, 0.7, 0.8]$. We run 5 experiments and

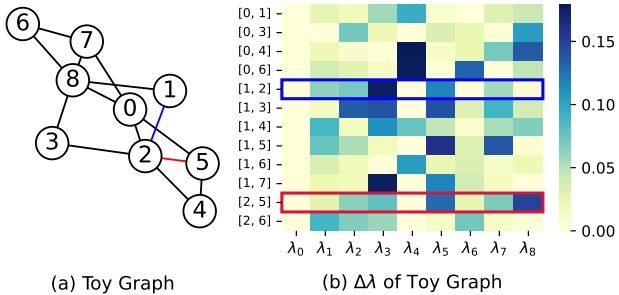

(a) Toy Graph      (b) $\Delta\lambda$ of Toy Graph

Figure 8: (a) Another toy graph $\mathcal{G}'$ consisting of nine nodes. (b) Absolute variation in eigenvalues of $\mathcal{G}'$ when adding an edge at diverse positions. The red and blue rectangles represent when adding the corresponding edges in $\mathcal{G}'$ and the change of the eigenvalues.

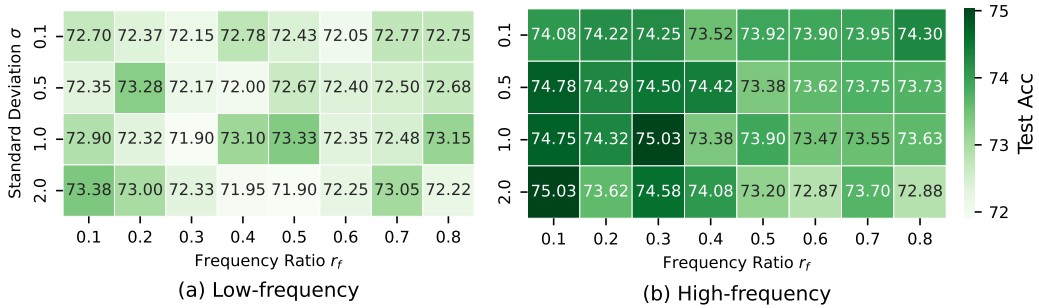

(a) Low-frequency           (b) High-frequency

Figure 9: Effects of different hyperparameter combinations on the IMDB-BINARY dataset in the supervised learning setting for graph classification via adding noise to (a) low-frequency and (b) high-frequency eigenvalues, respectively. The evaluation metric is accuracy.

report the average values in Figure 9. Our observations indicate that introducing noise in the high-frequency components tends to enhance the test accuracy more markedly than when infused in the low-frequency regions, as illustrated by the prevailing lighter color in Figure 9a. This observation resonates with the insights gleaned from Section 3. Building upon these general observations, we further elucidate the effects of each specific parameter below.

- **Effects of Standard Deviation $\sigma$.** For low-frequency components, the introduction of noise seems to not exhibit a consistent influence on accuracy. In contrast, when noise is applied to high-frequency components, we observe a discernible trend: accuracy tends to increase with increasing standard deviations. This suggests that the diversity introduced by elevating the standard deviation of noise can potentially bolster the classification performance of generated graphs.

- **Effects of Frequency Ratio $r_f$.** Similar to $\sigma$, for low-frequency components, increasing $r_f$ does not consistently enhance or degrade accuracy across different standard deviations. On the other hand, in the high-frequency regime, a subtle trend emerges. As $r_f$ increases, there is a nuanced shift in accuracy, suggesting that the spectrum of frequencies impacted by the noise has a nuanced interplay with the graph's inherent structures and the subsequent classification performance.

We also conduct hyperparameter analysis in different learning settings. Figure 10 shows performances across multiple datasets, which reveals distinct trends in performance related to augmentation probability ($aug_prob$) and frequency ratio ($aug_freq_ratio$). Specifically, for the DD dataset, performance peaks with a high $aug_freq_ratio$ and $aug_prob$, suggesting a preference for more frequent augmentations. In contrast, the MUTAG dataset shows optimal results at a lower frequency but higher probability, indicating a different augmentation response. The NCI1 dataset's best performance occurs at higher values of both parameters, while REDDIT-BINARY favors moderate to high frequency combined with a high probability, achieving its peak performance under these conditions.

These patterns highlight the necessity of customizing hyperparameters to each dataset for optimal augmentation effectiveness.

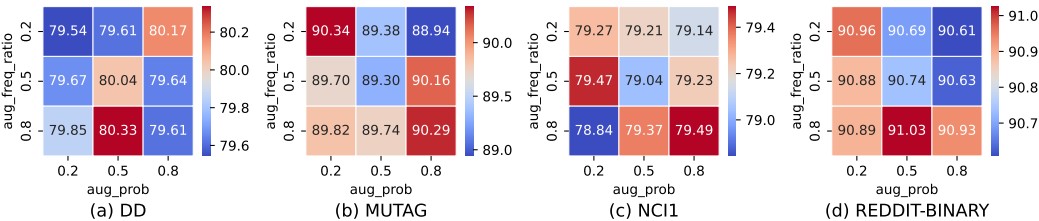

Figure 10: Effects of different hyperparameter combinations on different datasets in the unsupervised learning setting for graph classification via masking. The evaluation metric is accuracy.

**Complexity and Time Analysis.** Theoritically, the computational bottleneck of our data augmentation method primarily stems from the eigen-decomposition and reconstruction of the Laplacian matrix. For a graph with $n$ nodes, the computational complexity of both operations is $O(n^3)$. In terms of implementation, we have measured the time cost required by our method. For each $n$, we randomly generated 100 graphs and recorded the average time and standard deviation required for our data augmentation method. The results, presented in Table 8, are measured in milliseconds. The average number of nodes in commonly used graph classification datasets is approximately between 10 and 500. Therefore, in the majority of practical training scenarios, the average time consumption of our algorithm for augmenting a single graph is roughly between 1 millisecond and 40 milliseconds. The experiments conducted here did not employ any parallel computing or acceleration methods. However, in actual training processes, it is common to parallelize data preprocessing using multiple workers or to precompute and store the eigen-decomposition results of training data. Therefore, the actual time consumption required for our method in implementations will be even lower.

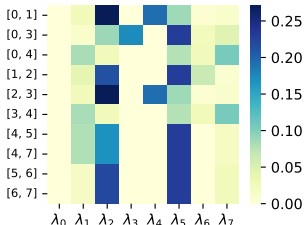

Figure 7: Absolute variation of eigenvalues when dropping different edges of the toy graph.

Table 8: Time cost required by our method. $n$: Number of nodes.

|          | $n = 10$      | $n = 20$      | $n = 100$     | $n = 200$     | $n = 500$      | $n = 1000$       |
| -------- | ------------- | ------------- | ------------- | ------------- | -------------- | ---------------- |
| Time(ms) | $0.76 \pm 0.55$ | $0.89 \pm 0.43$ | $2.50 \pm 0.50$ | $6.45 \pm 0.58$ | $41.35 \pm 2.39$ | $230.75 \pm 8.76$ |

# E   MORE DISCUSSIONS

**Intuitions & Advantages of proposed methods.** (1) *Properties Preservation.* Data augmentation should not only increase the quantity of training data but also enrich the quality of the learning experience for the model. Here, quality refers to the diversity, relevance, and realism of the augmented data. Therefore, property-retentive augmentations provide a more genuine learning context for the model, thus directly improving performance. (2) *Global Perspective.* Looking at the graph globally allows us to understand the larger structures and patterns within the graph. By making broader changes to the graph's structure, global augmentations can create more diverse training instances, compared to local augmentations which might only create minor variations of the existing instances, therefore enhancing understanding of complex graph relationships and contributing to improved performance.

**Broader Impact.** Through a spectral lens, our Dual-Prism (DP) augmentation method presents both significant advancements and implications in the realm of graph-based learning. This can lead to improved performance, robustness, and generalizability of graph neural networks (GNNs) across a myriad of applications, from social network analysis to molecular biology. In addition, by utilizing spectral properties, our method provides a more transparent approach to augmentation. This can help

researchers and practitioners better understand how alterations to graph structures impact learning outcomes, thereby aiding in the interpretability of graph data augmentation.

**Limitations & Future Directions.** A potential limitation of this study is its primary emphasis on homophily graphs. In contrast, heterophily graphs, where high-frequency information plays a more crucial role, are not extensively addressed (Bo et al., 2021). Looking ahead, it would be worth investigating learning strategies tailored to selectively alter eigenvalues, ensuring adaptability across diverse datasets. This includes developing methods to safely create realistic augmented graphs and experimenting with mix-up techniques involving eigenvalues from different graphs.

**Comparison with Existing Works.** There are two related works about the spectral view on graph data augmentation, i.e., (Liu et al., 2022; Lin et al., 2022), while both are grounded in the GCL framework. Specifically, Liu et al. (2022) proposes a rule to find the optimal contrastive pair under the GCL framework instead of a general augmentation method. GCL-SPAN Lin et al. (2022) centers on maximizing variance in the spectral domain, our observations indicate that overall spectral changes don't always align with graph properties, as detailed in Section 3. Thus, constraining specific eigenvalues to remain invariant might be a more effective strategy for generating valid augmented graphs. Despite GCL-SPAN (Lin et al., 2022) also utilizing a spectral perspective, it in fact still modifies the spatial domain while optimizing in the spectral realm. In contrast, our techniques directly make alterations in the spectral domain, leading to more meaningful and effective alterations. This is evident by our methods' superior performance in graph classification (see in Section 5.3) and underscores the efficiency of straightforward spectral modifications in creating more effective and discerning augmented graphs for graph classification. In addition, the DP method is specifically designed for graph classification tasks. Unlike node classification tasks Yoo et al. (2022) that emphasize node features and local structures, our approach is rooted in the analysis of global graph structures.

**Rationale for Choosing Graph Laplacian Decomposition**. In our methodology, we chose to decompose the graph Laplacian $L$ (where $L = D - A$) to do the perturbation and then reconstruct the graph. In terms of implementation, an alternative method can be directly decomposing $A$ and perturbing its smaller eigenvalues. However, our motivation for this work is more on the inherent properties of graphs, and $L$ offers a more nuanced reflection of these properties compared to $A$ Lutzeyer & Walden (2017). For future scenarios involving more complex disturbances to eigenvalues, leveraging the eigenvalues of $L$ would be a more appropriate approach. In addition, our decision to decompose $L$ also follows general spectral graph convolution methodologies Kipf & Welling (2016a).

