# OpenReview forum: "Through the Dual-Prism: A Spectral Perspective on Graph Data Augmentation for Graph Classification"
_ICLR.cc/2024/Conference — Submitted to ICLR 2024_

### Official Review · Reviewer_oQQa · 2023-10-18

**Soundness:** 2 fair
**Presentation:** 3 good
**Contribution:** 2 fair
**Rating:** 5
**Confidence:** 4

**Summary:**

This paper focuses on developing a more property-conserving and structure-sensitive augmentation method. To achieve this, authors first investigate the interplay between graph properties, their augmentation, and their spectral behavior to derive that keeping the low-frequency eigenvalues unchanged can preserve the critical properties at a large scale. They then propose Dual-Prism (DP), an augmentation method that adeptly retains essential graph properties while diversifying augmented graphs.

**Strengths:**

1. This paper is well-motivated. Developing a more property-conserving augmentation method has long been focused on.
2. Comprehensive experiments prove the performance of the proposed method.

**Weaknesses:**

1. It has long been proven that low-frequency information is valuable for graphs and the idea to augment more high-frequency components can be easily derived from previous works[1], which makes the proposal less innovative. What is the advantage of the proposed method in maintaining low-frequency information? The authors should also compare DP with SpCo[1] in the experiments.
2. The proposed method looks not efficient enough. It seems that Algorithm 1 involves eigenvalue decomposition and Laplacian Matrix reconstruction, which are both expensive. A time analysis would make the proposal more convincing.
3. The experiment currently lacks graphs with large node numbers such as ogbn-arxiv and ogbn-proteins[2].
4. Why is the improvement of the DP method over previous ones marginal in some cases in Tables 1 to 4?
5. In Obs 4, proof to the proposition "preserving key eigenvalues while modifying others enables the generation of augmented graphs that
uphold foundational properties"  is relatively insufficient, especially when the authors use the spectral variation defined only by a single previous work.
6. The font size in some figures is too small.

[1] Revisiting graph contrastive learning from the perspective of graph spectrum. Advances in Neural Information Processing Systems, 2022.\
[2] Open graph benchmark: Datasets for machine learning on graphs. Advances in Neural Information Processing Systems, 2020.

**Questions:**

See weaknesses.

---

> ### Author Response · Authors · 2023-11-19
>
> We thank the reviewer for the thorough evaluation and constructive feedback on our manuscript. We address each of the points raised below:
>
> **[Weaknesses]**
>
> - **W1.** Given the known value of low-frequency information in graphs [1], the proposal is less innovative. Compare with SpCo [1] in your experiments.
>
>     **A1.** Thank you for your insightful question. We will respond to your questions point by point. (1) Firstly, we would like to clarify that our proposed DP method is specifically designed for graph classification tasks. Unlike node classification tasks that emphasize node features and local structures, our approach is rooted in the analysis of global graph structures. (2) In terms of innovation, the DP method distinctly preserves low-frequency components, aiming to uphold the structural integrity of the graph's global property, which is a departure from previous methods. We also have thoroughly examined numerous related studies as detailed in [3,4] and determined that our method distinctly differs from other approaches. (3) Regarding the comparison with SpCo [1], our initial consideration included this comparison. However, SpCo primarily addresses node classification, focusing on local features and individual node importance. In contrast, our DP method is honed for graph classification, which necessitates a broader view, concentrating on the overall structure of the graph. This key difference in application scope and focus is the reason behind SpCo's exclusion from our comparative analysis. Your query has prompted us to enhance the discussion in Appendix E, providing a more in-depth explanation. We hope that this expanded discussion will offer a clearer understanding of our method's direction. Thank you.
>
> - **W2.** The proposed method looks not efficient enough. A time analysis would make the proposal more convincing.
>
>     **A2.** Thanks for pointing it out.  We acknowledge the significance of conducting a time and complexity analysis, which we have included in our revised manuscript. The primary computational demand of our data augmentation method arises from the eigen-decomposition and reconstruction of the Laplacian matrix. Theoretically, for a graph with $n$ nodes, these operations exhibit a computational complexity of $O(n^3)$. Practically, we evaluated the time efficiency of our method by generating 100 random graphs for each value of $n$ and recording the average time and standard deviation for augmentation, with the results detailed in milliseconds in the table below. The average node count in standard graph classification datasets typically ranges from 10 to 500. Consequently, in most real-world training scenarios, our algorithm's average time to augment a single graph falls between 1 millisecond and 40 milliseconds. These tests were conducted without the use of parallel computing or acceleration techniques. However, in actual deployment, parallelizing data preprocessing or pre-computing and storing the eigen-decomposition of training data is a common practice, potentially reducing the real-world time requirement of our method even further.
>
>
> |                     | Theoretical | $n=10$        | $n=20$        | $n=100$       | $n=200$       | $n=500$        | $n=1000$        |
> |---------------------|-------------|---------------|---------------|---------------|---------------|----------------|-----------------|
> | Complexity/Time(ms) | $O(n^3)$    | $0.76\pm0.55$ | $0.89\pm0.43$ | $2.50\pm0.50$ | $6.45\pm0.58$ | $41.35\pm2.39$ | $230.75\pm8.76$ |

---

> ### Author Response · Authors · 2023-11-19
>
> - **W3.** The experiment currently lacks graphs with large node numbers such as ogbn-arxiv and ogbn-proteins[2].
>
>     **A3.** Thank you for your suggestions. In response, we'd like to offer the following clarifications: (1) The datasets 'ogbn-arxiv' and 'ogbn-proteins' you referred to are tailored for node classification. Our method, however, is specifically designed for graph classification. Therefore, our experimental focus has been on datasets relevant to graph classification tasks. (2) Regarding your concern about experiments on the large dataset, We have conducted experiments on the 'ogbg-molhiv' dataset, which we consider significant due to its size and complexity for graph classification. This dataset contains $41,127$ graphs, with an average of $25.5$ nodes per graph, surpassing others like REDDIT-BINARY in scale. Detailed statistics of these datasets are provided in Table 5 of our manuscript. (3) To further address your concern about evaluation on larger datasets, we have extended our experiments to include another larger dataset, 'ogbg-ppa', containing $158,100$ graphs and the number of nodes per graph is 243.4, which is the largest graph in 'ogbg' dataset in [2]. The experiment is conducted using GIN network and also shows the performance of S-Mixup [7] as a baseline, which is presented in the table below. Due to the time limitation and the extensive scale of the dataset, we were unable to complete experiments with other baselines. We commit to completing all baselines and will include these results in our future version. We hope this addresses your concern regarding dataset size and applicability.
>
>     |     S-Mixup     |      DP_Noise     |      DP_Mask      |
>     |:--------------:|:--------------:|:--------------:|
>     | 19.21$\pm$0.59 | 21.79$\pm$0.03 | 22.04$\pm$0.08 |
>
> - **W4.** Why is the improvement of the DP method over previous ones marginal in some cases in Tables 1 to 4?
>
>     **A4.** Thank you for your question. We have included this point in our original manuscript, which we can further clarify here: **(a) Inherent Model and Dataset Characteristics**: The slight improvements in supervised learning settings can be attributed to the inherent characteristics of GIN and GCN models, as well as the specific nature of the datasets. GIN is known for its precision in delineating complex structural intricacies of graphs, while GCN is characterized by its smoothing effect. Our methods’ superiority in diversifying the graphs’ structures naturally amplifies GIN’s strengths. In contrast, GCN may not be as adept at leveraging the enhancements offered by our augmentation techniques. The divergence is evident in our REDD-B dataset results, where GCN, combined with our method, shows limited improvement. **(b) Context-Dependent Performance**: The efficacy of DP methods varies depending on the learning context. In supervised and unsupervised learning settings (as in Tables 1 and 2), the improvements are more pronounced, especially when dealing with complex graph structures (e.g., NCI1). In contrast, in semi-supervised and transfer learning contexts (as reflected in Tables 3 and 4), the improvements, while still present, are less dramatic. This variance is partly due to the different challenges and requirements of each learning paradigm.
>
> - **W5.** The proof for Ob4 is not enough, especially when the authors use the spectral variation defined only by a single previous work.
>
>     **A5.** Thank you for your feedback. In response to your concerns about Obs4, we have enhanced our demonstration by introducing an additional property, the Average Shortest Path Length (ASPL), showcased in Figure 3, which indicates that ASPL further supports our observations regarding the correlation between graph properties and specific eigenvalues, hopefully addressing your concerns. Regarding the measurement of spectral variation, we acknowledge the initial oversight in our referencing. To rectify this, we have now included additional references [5, 6], which also employ Frobenius distance or $L_2$ distance for spectral analysis, thus providing a more comprehensive theoretical basis for it.

---

> ### Author Response · Authors · 2023-11-19
>
> - **W6.** The font size in some figures is too small.
>
>      **A6.** Thank you for pointing out the issue. We have updated Figure 1 in the revised manuscript to ensure that the text is larger and more legible, enhancing readability and clarity. We are open to any additional suggestions you might have regarding other figures and will promptly make necessary adjustments as required. Thank you!
>
>
> Thank you again for your detailed review and insightful feedback. We have meticulously addressed and incorporated your suggestions into our revised manuscript. **We believe these are not essentially technical issues.** Considering the other reviewers agree that our paper has good merits such as satisfactory novelty and comprehensive evaluation, we sincerely hope that you could reconsider our score. Thank you so much!
>
> **Reference:**
>
> [1] Revisiting graph contrastive learning from the perspective of graph spectrum. Advances in Neural Information Processing Systems, 2022.
>
> [2] Open graph benchmark: Datasets for machine learning on graphs. Advances in Neural Information Processing Systems, 2020.
>
> [3] Data augmentation for deep graph learning: A survey. KDD. 2022.
>
> [4] Graph data augmentation for graph machine learning: A survey. arXiv. 2022.
>
> [5] Augmentation-free graph contrastive learning with performance guarantee. arXiv. 2022.
>
> [6] Metrics for graph comparison: a practitioner's guide. 2020.
>
> [7] Graph mixup with soft alignments. ICML. 2023.

---

> > ### Comment · Reviewer_oQQa · 2023-11-20
> > **Thank you for the response.**
> >
> > Thank you for the detailed response. I have updated my score.

---

> > > ### Author Response · Authors · 2023-11-20
> > >
> > > We deeply appreciate your willingness to re-evaluate our work! Thanks so much for your thoughtful and constructive feedback throughout this process, as it has been instrumental in enhancing the quality of our work. Thank you!

---

### Official Review · Reviewer_7Xfr · 2023-10-22

**Soundness:** 3 good
**Presentation:** 4 excellent
**Contribution:** 2 fair
**Rating:** 5
**Confidence:** 3

**Summary:**

This paper studies the graph-level tasks with graph augmentation techniques. To be specific, they propose to perturb the high-frequency part of the given graphs to generate augmented graph samples, so as to boost the performance of graph-level tasks.

**Strengths:**

S1. The presentation of this paper is excellent, and the paper is well-organized.

S2. This paper includes comprehensive experiments, including supervised, unsupervised, and transfer learning settings.

S3. The proposed method is concise but its performance on the supervised learning tasks is good.

**Weaknesses:**

W1. The main concern of this paper is its novelty, which is low and being studied in many existing works.

W2. A minor drawback of this paper is its performance. It shows strong performance in the supervised settings but gets average performance in other settings. In addition, some experimental results are missing, which is not expected.

I will elaborate more in detail in the Questions setting.

**Questions:**

Q1. My main concern with this paper is its novelty. Which shares great overlap with this paper [1], as multiply mentioned by the authors. Though they are not invented for the same purpose, it is not hard to transfer the idea from [1] into the context of this paper.

Q2. In section 3.2, many observations have been mentioned by existing works. For example, **Obs 2. Low-frequency components display greater resilience to edge alterations** has been mentioned in existing work [2]. **Obs 4. Specific low-frequency eigenvalues are
closely tied to crucial graph properties.**, as this paper mentioned in Section 4.3, has been studied thoroughly by Chung in the spectral graph theory [3].

Q3. The proposed Algorithm 1 first decomposes the graph Laplacian L,  perturbs the high-frequency part (larger eigenvalues of L), and finally reconstructs the perturbed adjacency matrix. I think a simpler version is directly decomposing the adjacency matrix A and perturbing its (A's) small eigenvalues. From this perspective, it is similar to many classic low-rank approximation-based works on graphs.

Q4. The performance in the supervised setting is good, which is shown in Table 1. However, its performance in unsupervised learning (Table 3) and transfer learning settings (Table 4) is average.

Q5. A suggestion for this paper is to finish experiments in Tables 1,2, and 3, where now they are shown '-'. Ideally, if the experiments are not conducted in existing papers, authors should implement the baseline methods and report the results in those missing setting by themselves.

[1] Lin, Lu, Jinghui Chen, and Hongning Wang. "Spectral Augmentation for Self-Supervised Learning on Graphs." In The Eleventh International Conference on Learning Representations. 2023.

[2] Wang, Haonan, Jieyu Zhang, Qi Zhu, and Wei Huang. "Augmentation-free graph contrastive learning with performance guarantee." arXiv preprint arXiv:2204.04874 (2022).

[3] https://mathweb.ucsd.edu/~fan/mypaps/fanpap/111diameters.pdf

---

> ### Author Response · Authors · 2023-11-19
>
> Thank you for your thorough review and constructive feedback on our paper. We appreciate the time you've taken to provide valuable insights. We appreciate your acknowledgment of the originality, quality, and clarity of our work. Please find our responses below.
>
> **[Weaknesses]**
>
> Please see the Questions section.
>
> **[Questions]**
>
> - **Q1.** My main concern with this paper is its novelty. Which shares great overlap with this paper [1].
>
>     **A1.** We appreciate your question. As discussed in Appendix E, our approach, while superficially similar to [1] in utilizing the spectral domain, diverges significantly in both intent and execution. Specifically, the primary focus of [1] is on maximizing variance within the spectral domain, but our work reveals that such an approach does not consistently align with the preservation of graph properties. This led us to a different strategy: selectively constraining specific eigenvalues to remain constant, thus ensuring the generation of augmented graphs that maintain key properties. Furthermore, the methodology in [1] essentially alters the spatial domain, albeit within a spectral framework. In contrast, our technique directly intervenes in the spectral domain, resulting in modifications that are more targeted and consequential for graph augmentation. Therefore, we argue that while there is a superficial overlap in terms of using "spectrum" for "graph data augmentation", the core methodologies and outcomes are distinctly different. It is not sufficient to equate our approach with [1] merely based on the shared perspective of spectrum in graph data augmentation. To enhance clarity, we have added a table as follows. *We believe it should be a new direction and open up new avenues for spectral-based augmentation techniques, allowing more diverse research on doing the augmentation on the spectrum.*
>
> | **Aspect**                      | **Our Approach**                                                | **Approach in [1]**                                             |
> |---------------------------------|-----------------------------------------------------------------|-----------------------------------------------------------------|
> | **Primary Focus**               | Preserving graph properties by selectively constraining specific eigenvalues. | Maximizing variance within the spectral domain.                 |
> | **Impact on Graph Properties**  | Ensures preservation of key graph properties.                   | Does not consistently align with preservation of graph properties. |
> | **Methodology**                 | Direct intervention in the spectral domain for targeted modifications. | Alters the spatial domain within a spectral framework.          |
> | **Outcome**                     | Generation of augmented graphs that maintain key properties.    | Modifications that may not be as targeted for graph augmentation. |
> | **Overall Approach**            | Represents a distinct direction, emphasizing spectral-based augmentation with a focus on property preservation. | Focuses on spectral variance, potentially overlooking the impact on graph properties. |
>
>
>
> - **Q2.** Some observations have been mentioned in existing works.
>
>   **A2.** Thank you for highlighting this. We would like to clarify that our intention in including these observations was to provide foundational context for our methodology, rather than to claim these as novel findings. While it is true that these concepts have been explored in previous studies, our work aims to apply these established theories in a new framework. We are grateful for drawing our attention to [2] and [3] that we may have initially overlooked, which have been thoughtfully integrated into the revised version of our paper to enrich and complement our research.
>
> - **Q3.** I think a simpler version is directly decomposing the adjacency matrix A and perturbing its (A's) small eigenvalues.
>
>     **A3.** Thank you for your thoughtful suggestion. We acknowledge that, from an implementation standpoint, these two methods are essentially equivalent. However, our motivation is more on the inherent properties of graphs, and $L$ offers a more nuanced reflection of these properties compared to $A$ [4]. For future scenarios involving more complex disturbances to eigenvalues, leveraging the eigenvalues of $L$ would be a more appropriate approach. In addition, our decision to decompose $L$ is also following general spectral graph convolution methodologies [5]. We greatly value your feedback and have included a comprehensive discussion on this topic in Appendix E for additional clarification.

---

> ### Author Response · Authors · 2023-11-19
>
> - **Q4.** The performance in the supervised setting is good, while its performance in unsupervised learning and transfer learning settings is average.
>
>     **A4.** Thank you for your comments. We understand the concern that our method may not achieve SOTA on all datasets in all settings. We add the student T-Test in Tables 1-4 to clarify the significance level of improvement. Yet, it's important to highlight that the essence of our research lies not solely in the results but in a new path for graph data augmentation through eigenvalue manipulation, which lays the groundwork for developing more sophisticated techniques. We believe future works can design improvement methods to strategically select eigenvalues and calibrate augmentation levels. As the method evolves, we anticipate that it will not only match but potentially exceed its current efficacy in supervised settings.
>
> - **Q5.** A suggestion is to finish experiments in Tables 1,2, and 3, where now they are shown '-'.
>
>     **A5.** Thank you for your valuable suggestion. We understand that completing the missing baseline experiments would enhance the demonstration of our method's performance. We have devoted considerable effort to complete these experiments. However, ensuring the comparison consistency required us to reimplement every missing method, a task we found challenging to complete within the limited time of the discussion phase. Given that the missing baselines constitute only a minor portion of our study, we hope the comparisons with the presented baseline results sufficiently demonstrate the advantages of our method. We commit to completing and presenting all missing experimental results in future revisions. Thanks for your suggestion.
>
> Thank you again for your valuable feedback. We have revised our manuscript based on your feedback. **We believe these are not essentially technical issues,** and we sincerely hope that you could reconsider our score. Thank you so much!
>
> **Reference:**
>
> [1] Lin, Lu, Jinghui Chen, and Hongning Wang. "Spectral Augmentation for Self-Supervised Learning on Graphs." In The Eleventh International Conference on Learning Representations. 2023.
>
> [2] Wang, Haonan, Jieyu Zhang, Qi Zhu, and Wei Huang. "Augmentation-free graph contrastive learning with performance guarantee." arXiv preprint arXiv:2204.04874 (2022).
>
> [3] https://mathweb.ucsd.edu/~fan/mypaps/fanpap/111diameters.pdf
>
> [4] Comparing graph spectra of adjacency and laplacian matrices. arXiv preprint arXiv:1712.03769, (2017)
>
> [5] Semi-supervised classification with graph convolutional networks. ICLR. (2017)

---

> ### Author Response · Authors · 2023-11-22
>
> Dear Reviewer 7Xfr,
>
> Thank you once again for your insightful feedback on our manuscript. **We have carefully considered your comments and made corresponding revisions to our manuscript.** Hope these can address your concern. We would like to kindly remind you that the discussion period concludes on **November 22, 2023**. We are eager to engage in a dialogue about your remarks and further enhance our work. Your expertise and perspective are greatly valued, and we look forward to a productive discussion.
>
> Best regards,
> Authors

---

> > ### Comment · Reviewer_7Xfr · 2023-11-22
> >
> > I thank the authors for their rebuttal. I have read all the responses and I think my concerns about the novelty and many missing experiments are not solved (to be frank, they are hard to improve during the short rebuttal phase). In my view, the current version of this paper is still a bit below the borderline. I prefer to keep my original evaluation.

---

> > > ### Author Response · Authors · 2023-11-22
> > >
> > > We sincerely thank you for your time and comments. We understand your decision to maintain your original evaluation, but we are open to engaging in further dialogue to address and resolve these points. Thank you once again for your thoughtful review.

---

### Official Review · Reviewer_48h8 · 2023-10-26

**Soundness:** 4 excellent
**Presentation:** 3 good
**Contribution:** 3 good
**Rating:** 6
**Confidence:** 4

**Summary:**

Graph neural networks (GNNs) have become the preferred tool to process graph data. This paper aims to develop property-conserving and structure-sensitive augmentation methods. Through a spectral lens, the authors investigate the interplay between graph properties, their augmentation, and their spectral behavior, and found that keeping the low-frequency eigenvalues unchanged can preserve the critical properties at a large scale when generating augmented graphs.

**Strengths:**

1. The writing is clear, and the paper is easy to follow.
2. Instead of proposing another random approach, the authors provide their rationale clearly and comprehensively based on empirical evidence.
3. The experiments are done extensively for 4 different tasks, on 21 datasets, and against numerous competitors.

**Weaknesses:**

1. Since the accuracy improvement over competitors is not dramatic, statistical tests such as the Wilcoxon signed-rank test would be beneficial.
2. Changing high-frequency eigenvalues is similar to making small, marginal changes to the graph structure while preserving the core properties, such as connectivity. In that sense, NodeSam [1] and MotifSwap [2] are better competitors than mixup-based approaches, which induce more changes to the structure.
3. Although this paper discusses extensively the reasons why we should focus on high-frequency eigenvalues, there is little discussion on how to actually modify them. Simply using random masking or adding random noise appears too naive. Additional discussion on this part, e.g., how to safely alter these eigenvalues to create plausible augmented graphs, or how we might mix-up the eigenvalues between different graphs, would be valuable.

[1] J. Yoo et al. “Model-Agnostic Augmentation for Accurate Graph Classification.” WWW 2022

[2] J. Zhou et al. "Data Augmentation for Graph Classification.” CIKM 2020

**Questions:**

1. How long does it take to eigendecompose the matrix L? Is the complexity linear with the size of a graph?
2. Apart from Figure 2b, could you provide more examples of augmented graphs resulting from changes to the eigenvalues?

---

> ### Author Response · Authors · 2023-11-19
>
> Thank you for your detailed review of our submission. We appreciate your recognition of the clarity of our writing, the rationale behind our approach, and the extensive experimental validation. Below, we address your comments in a point-by-point manner.
>
> **[Weaknesses]**
>
> - **W1.** Statistical tests would be beneficial.
>
>     **A1.** Thanks for your suggestion. We agree that incorporating statistical tests could provide a more robust validation of our results. In response to your feedback, we have included the student's t-test results in our revision (see * and ** in Table 1-4). We sincerely hope this addresses your concern. Thank you!
>
> - **W2.** NodeSam and MotifSwap are better competitors than mixup-based approaches in terms of marginal changes to the graph structure.
>
>     **A2.** Thank you for this constructive comment. We have thoroughly reviewed the papers (i.e., NodeSam and MotifSwap) and appreciated their significant contributions in graph data augmentation. In line with your suggestions, we intend to carry out additional experiments involving these two methods. Regrettably, time constraints have prevented their inclusion in the current version. However, we are committed to expanding our experiments and incorporating these results in the final version of our manuscript. Thank you for guiding this important aspect of our research.
>
> - **W3.** The paper simply using random masking or adding random noise appears naive. Additional discussion on this part, e.g., how to safely alter these eigenvalues to create plausible augmented graphs, or how we might mix-up the eigenvalues between different graphs, would be valuable.
>
>     **A3.** We appreciate your insightful comments. In this work, our DP method stands out as a concise yet efficient approach, as highlighted in Reviewer 7Xfr' strengths section. Despite its simplicity, this method effectively preserves the graph's properties, as demonstrated in our comparison with other randomly-mannered methods in the introduction. The effectiveness of our approach is further evidenced by the performance results in our experiments section. Meanwhile, we agree with your suggestion that exploring more sophisticated strategies is a promising direction for our future research. To this end, we have broadened the discussion in Appendix E to encompass methods for safely creating plausible augmented graphs in our revised manuscript. Thanks!
>
> **[Questions]**
>
> - **Q1.** How long does it take to eigendecompose the matrix L? Is the complexity linear with the size of a graph?
>
>     **A1.** Thanks for pointing out this question. We agree on the importance of time and complexity analysis and thus provide it as follows (also included in our revision). Theoretically, the computational bottleneck of our data augmentation method primarily stems from the eigen-decomposition and reconstruction of the Laplacian matrix. For a graph with $n$ nodes, the computational complexity of both operations is $O(n^3)$. In terms of implementation, we have measured the time cost required by our method. For each $n$, we randomly generated $100$ graphs and recorded the average time and standard deviation required for our data augmentation method. The results, presented in the table below, are measured in milliseconds. The average number of nodes in commonly used graph classification datasets is approximately between $10$ and $500$. Therefore, in the majority of practical training scenarios, the average time consumption of our algorithm for augmenting a single graph is roughly between $1$ millisecond and $40$ milliseconds. The experiments conducted here did not employ any parallel computing or acceleration methods. However, in actual training processes, it is common to parallelize data preprocessing using multiple workers or to pre-compute and store the eigen-decomposition results of training data. Therefore, the actual time consumption required for our method in implementations will be even lower.
>
> |                     | Theoretical | $n=10$        | $n=20$        | $n=100$       | $n=200$       | $n=500$        | $n=1000$        |
> |---------------------|-------------|---------------|---------------|---------------|---------------|----------------|-----------------|
> | Complexity/Time(ms) | $O(n^3)$    | $0.76\pm0.55$ | $0.89\pm0.43$ | $2.50\pm0.50$ | $6.45\pm0.58$ | $41.35\pm2.39$ | $230.75\pm8.76$ |

---

> ### Author Response · Authors · 2023-11-19
>
> - **Q2.** Apart from Figure 2b, could you provide more examples of augmented graphs resulting from changes to the eigenvalues?
>
>     **A2.** We appreciate your inquiry. To address your concern, we have included a supplementary example consisting of a nine-node toy graph, which effectively demonstrates the same results and principles discussed in our main analysis. We have presented it in **Appendix D, Figure 8** of our revised manuscript, where we showcase its alignment with our primary observations and conclusions. We believe that they address your concern. Thank you for your attention!
>
> Thank you again for your insightful comments. We have addressed and incorporated your suggestions into our revised manuscript.

---

> > ### Comment · Reviewer_48h8 · 2023-11-22
> > **Reply to Authors**
> >
> > Thank you for the detailed response. My concerns have been addressed well. I will maintain my positive score.

---

> > > ### Author Response · Authors · 2023-11-22
> > >
> > > Thanks so much for your invaluable comments and suggestions throughout this process, which have enhanced the quality of our work. Thank you!

---

### Official Review · Reviewer_oF9L · 2023-11-06

**Soundness:** 3 good
**Presentation:** 3 good
**Contribution:** 4 excellent
**Rating:** 8
**Confidence:** 4

**Summary:**

This paper introduces a novel graph data augmentation method called Dual-Prism (DP), which aims to retain essential graph properties while diversifying augmented graphs. The authors draw inspiration from the way prisms decompose and reconstruct light and how polarizers selectively filter light to design their own "polarizer". They conduct extensive experiments on diverse real-world datasets and demonstrate that their proposed methods can achieve state-of-the-art performance on most of the datasets. This work provides a promising new direction for graph data augmentation.

**Strengths:**

The Dual-Prism (DP) augmentation method proposed in this paper is a novel approach to graph data augmentation. The authors draw inspiration from optics to design their own "polarizer" that retains essential graph properties while diversifying augmented graphs. This innovative approach provides a new direction for graph data augmentation.

The authors conduct extensive experiments on 21 real-world datasets spanning various learning paradigms. The experimental results demonstrate that their proposed methods can achieve state-of-the-art performance on most of the datasets. This extensive evaluation provides strong evidence for the efficacy of the DP augmentation method.

The authors provide empirical evidence to substantiate their approach. They explain the rationale behind their DP method and how it skillfully preserves graph properties while ensuring diversity in augmented graphs. This work also proposes the globally-aware and property-retentive augmentation methods, DP-Noise and DP-Mask, which are able to preserve inherent graph properties while simultaneously enhancing the diversity of augmented graphs.

**Weaknesses:**

The authors could delve deeper into the influence of various hyperparameters on the performance of the Dynamic Programming (DP) method. Although they provide some details on the hyperparameters used in their experiments, a more detailed exploration could potentially identify optimal hyperparameters for different types of graphs and learning tasks.

Moreover, it would be compelling to examine the effectiveness of the proposed DP method on larger and more complex graphs. Despite conducting experiments on 21 real-world datasets, extending this to larger, more complex graphs could further validate the efficiency of their proposed method and offer valuable insights into its scalability.

**Questions:**

The authors could enhance their study by further investigating the effect of various hyperparameters on the Dynamic Programming (DP) method's performance. While details of the used hyperparameters are given, a more comprehensive exploration could help identify optimal hyperparameters for diverse graph types and learning tasks. Additionally, testing the proposed DP method on larger and more complex graphs, beyond their 21 real-world datasets, could further validate the method's efficiency and provide insights into its scalability.

---

> ### Author Response · Authors · 2023-11-19
>
> We greatly appreciate your positive feedback on the novelty and effectiveness of the DP method and our extensive experimentation on a variety of real-world datasets. Your feedback is invaluable and helps enhance the quality of our paper. Below, we respond to your comments point-by-point.
>
> **[Weaknesses]**
>
> - **W1.** The authors can more thoroughly investigate how different hyperparameters impact the performance of the DP method.
>
>     **A1.** Thanks for your invaluable suggestion. We agree that a more detailed exploration of the influence of hyperparameters is important. We have conducted hyperparameter analysis on unsupervised learning settings (see below table). We also included them in **Appendix D, Figure 10** of our revised version. Specifically, for the DD dataset, performance peaks with a high aug_freq_ratio and aug_prob, suggesting a preference for more frequent augmentations. In contrast, the MUTAG dataset shows optimal results at a lower frequency but higher probability, indicating a different augmentation response. The NCI1 dataset's best performance occurs at higher values of both parameters, while REDDIT-BINARY favors moderate to high frequency combined with a high probability, achieving its peak performance under these conditions. These patterns highlight the necessity of customizing hyperparameters to each dataset for optimal augmentation effectiveness.
>
> | Dataset       | aug_freq_ratio | aug_prob 0.20 | aug_prob 0.50 | aug_prob 0.80 |
> |---------------|----------------|---------------|---------------|---------------|
> | DD            | 0.2            | 79.54         | 79.61         | 80.17         |
> | DD            | 0.5            | 79.67         | 80.04         | 79.64         |
> | DD            | 0.8            | 79.85         | 80.33         | 79.61         |
> | MUTAG         | 0.2            | 90.34         | 89.38         | 88.94         |
> | MUTAG         | 0.5            | 89.70         | 89.30         | 90.16         |
> | MUTAG         | 0.8            | 89.82         | 89.74         | 90.29         |
> | NCI1          | 0.2            | 79.27         | 79.21         | 79.14         |
> | NCI1          | 0.5            | 79.47         | 79.04         | 79.23         |
> | NCI1          | 0.8            | 78.84         | 79.37         | 79.49         |
> | REDDIT-BINARY | 0.2            | 90.96         | 90.69         | 90.61         |
> | REDDIT-BINARY | 0.5            | 90.88         | 90.74         | 90.63         |
> | REDDIT-BINARY | 0.8            | 90.89         | 91.03         | 90.93         |
>
> - **W2.** Experiments on larger graphs could further validate the efficiency of the proposed method.
>
>     **A2.** Thank you for your suggestion. We understand the significance of such validation for assessing scalability and efficiency. We have indeed conducted experiments on the 'ogbg-molhiv' dataset, which encompasses 41,127 graphs, with an average node count of 25.5 per graph. Additionally, we have included experiments on the 'ogbg-ppa' dataset to further address your concern, which is an even larger dataset with 158,100 graphs in total and 243.4 edges per graph on average. The experiment is conducted using GIN network and also shows the performance of S-Mixup [2] as a baseline, which is presented in the table below. Due to the time limitation and the extensive scale of the dataset, we were unable to complete experiments with other baselines. We commit to completing all baselines and will include these results in our future version.  We hope this expanded experimental scope effectively addresses your concerns about evaluating our method's effectiveness on larger, complex graph datasets.
>
>
> |     S-Mixup [2]     |      Noise     |      Mask      |
> |:--------------:|:--------------:|:--------------:|
> | 19.21 ± 0.59 | 21.79 ± 0.03 | 22.04 ± 0.08 |
>
>
> **[Questions]**
>
> Please see the Weakness section.
>
> Again, we are grateful for your insightful comments and questions, which guide us in improving our work. Thank you!
>
> **Reference:**
>
> [1] Open graph benchmark: Datasets for machine learning on graphs. NeurIPS, (2020).
>
> [2] Graph mixup with soft alignments. ICML. (2023).

---

### Author Response · Authors · 2023-11-19

Dear Reviewers,

We would like to express our sincere gratitude to all the reviewers for their thorough evaluation and constructive feedback on our manuscript. Your insights have been invaluable in enhancing the quality and clarity of our work. We have made several revisions highlighted in red in our rebuttal version accordingly to address the concerns raised:

- **Enhanced Clarifications**: We have enhanced our manuscript with more in-depth explanations where necessary, such as the rationale for choosing graph laplacian decomposition and the future directions.

- **Complexity Analysis**: According to the reviewers' suggestion,  due to the computational intensity of eigendecomposition in our method, we have included a subsection to discuss the complexity and time analysis in Appendix D.

- **Additional Experiments**: Based on the feedback, we have conducted additional experiments to further validate the effectiveness of our proposed model on a larger graph dataset 'ogbg-ppa'. We also include more ablation analysis focusing on hyperparameters under different settings (i.e., unsupervised learning). We also have added some more empirical studies to prove our observations, e.g., Figure 3 and Figure 8.

- **Expanded Literature Discussion**: We have expanded our discussion on related works [1-5]. This helps in positioning our work in the broader context of the field.

- **Enhanced Figure Readability**: We have enhanced the quality of our figures by increasing the font size, ensuring they are more reader-friendly and easier to interpret.

We believe that these revisions have significantly improved our manuscript. We hope that our responses and the changes made address the concerns of the reviewers adequately.

Once again, thank you for your time and effort in reviewing our work. We look forward to your continued feedback.

Best regards,

Authors

**Reference**

[1] Haonan Wang. Augmentation-free graph contrastive learning with performance guarantee. arXiv. 2022.

[2] Peter Wills. Metrics for graph comparison: a practitioner’s guide. Plos one. 2020.

[3] Jiajun Zhou. Data augmentation for graph classification. CIKM. 2020.

[4] Kaize Ding. Data augmentation for deep graph learning: A survey. KDD. 2022.

[5] Tong Zhao. Graph data augmentation for graph machine learning: A survey. arxiv. 2022.

---

### Author Response · Authors · 2023-11-21
**A kind reminder about author-reviewer discussion deadline (Nov. 22nd)**

Dear Reviewers,

Thank you once again for your invaluable comments. We have revised our manuscript correspondingly, with changes highlighted in red for easy reference. We would like to gently remind you that the dicussion period will end on 22nd Nov. 2023. We were hoping to engage in some discussion regarding your comments. Thank you again and looking forward to discussing!

Best regards,
Authors

---

### Meta-Review · Area_Chair_pSQu · 2023-12-14

**Metareview:**

This paper addresses the persistent challenges of graph property distortions and limited structural changes in augmentation methods. Toward this, the authors propose the Dual-Prism (DP) augmentation method, incorporating DP-Noise and DP-Mask, which preserves critical graph properties by maintaining unchanged low-frequency eigenvalues. The authors argue that through extensive experiments, the efficiency of this approach is validated, offering a promising direction for graph data augmentation.

Four reviewers evaluated the paper, with two recommending acceptance and two suggesting rejection. Most reviewers agreed that the paper is well-written, clear, and easy to follow. They also acknowledged the strength of the paper in extensively evaluating 21 datasets across supervised, semi-supervised, unsupervised, and transfer learning settings. However, this AC and two reviewers expressed significant concern about the limited novelty of the paper, making acceptance challenging. While the authors' focus and intent may differ, the idea itself of preserving low-frequency components while augmenting high-frequency ones has been extensively studied, and the proposed method seemed to overlap with existing research, as pointed by the reviewers. The paper failed to convincingly demonstrate the advantages or differences of their approach in augmenting high-frequency components while maintaining low-frequency components. Following an internal discussion between the reviewers and the AC, as no reviewer championed the acceptance of the paper, the final decision was made to reject it.

**Justification For Why Not Higher Score:**

I cannot claim 100% certainty in my decision as I haven't kept up with the latest research in this field. However, based on my knowledge, I share the same concerns as the two negative reviewers regarding the lack of novelty in this paper. Additionally, while experiments were conducted in various settings, I find the results not particularly impressive. In most cases, the proposed method seems comparable to existing approaches or slightly superior within the confidence interval, making it challenging to conclusively assert its superiority. It would be nice if SAC can confirm the final decision.

**Justification For Why Not Lower Score:**

n/a

---

### Decision · Program_Chairs · 2024-01-16

Reject